# Modelling last glacial cycle ice dynamics in the Alps

Julien Seguinot[1,2], Susan Ivy-Ochs[3], Guillaume Jouvet[1], Matthias Huss[1], Martin Funk[1], and
Frank Preusser[4]

[1]Laboratory of Hydraulics, Hydrology and Glaciology, ETH Zürich, Switzerland
[2]Arctic Research Center, Hokkaido University, Sapporo, Japan
[3]Laboratory of Ion Beam Physics, ETH Zürich, Switzerland
[4]Institute of Earth and Environmental Sciences, University of Freiburg, Germany

*Correspondence to:* J. Seguinot (seguinot@vaw.baug.ethz.ch)

**Abstract.**

The European Alps, cradle of pioneering glacial studies, are one of the regions where geological markers of past glaciations are most abundant and well-studied. Such conditions make the region ideal for testing numerical glacier models based on simplified ice flow physics against field-based reconstructions, and vice-versa.

Here, we use the Parallel Ice Sheet Model (PISM) to model the entire last glacial cycle (120–0 ka) in the Alps, using horizontal resolutions of 2 and 1 km. Climate forcing is derived using present-day climate data from WorldClim and the ERA-Interim reanalysis, and time-dependent temperature offsets from multiple palaeo-climate proxies, among which only the EPICA ice core record yields glaciation during marine oxygen isotope stages 4 (69–62 ka) and 2 (34–18 ka) spatially and temporally consistent with the geological reconstructions, while the other records used result in excessive early glacial cycle ice cover and a late Last Glacial Maximum. Despite the low variability of this Antarctic-based climate forcing, our simulation depicts a highly dynamic ice sheet, showing that Alpine glaciers may have advanced many times over the foreland during the last glacial cycle. Ice flow patterns during peak glaciation are largely governed by subglacial topography but include occasional transfluences through the mountain passes. Modelled maximum ice surface is in average 861 m higher than observed trimline elevations in the upper Rhone Valley, yet our simulation predicts little erosion at high elevation due to cold-based ice. Finally, despite the uniform climate forcing, differences in glacier catchment hypsometry produce a time-transgressive Last Glacial Maximum advance, with some glaciers reaching their modelled maximum extent as early as 27 ka, and others as late as 21 ka.

## 1 Introduction

For nearly 300 years, montane people and early explorers of the European Alps learned to read the geomorphological imprint left by glaciers in the landscape, and to understand that glaciers had once been more extensive than today (e.g., Windham and Martel, 1744, p. 21). Contemporaneously, it was also observed in the Alps that glaciers move by a combination of meltwater-induced *sliding* at the base (de Saussure, 1779, §532), and viscous *deformation* within the ice body (Forbes, 1846). As glaciers flow and slide across their bed, they transport rock debris and erode the landscape, thereby leaving geomorphological traces of their former presence. In the mid-nineteenth century, more systematic studies of glacial features showed that Alpine glaciers

extended well outside their current margins (Venetz, 1821) and even onto the Alpine foreland (Charpentier, 1841), yielding the idea that, under colder temperatures, expansive *ice sheets* had once covered much of Europe and North America (Agassiz, 1840).

However, this glacial theory did not gain general acceptance until the discovery and exploration of the two present-day ice
sheets on Earth, the Greenland and Antarctic ice sheets, provided a modern analogue for the proposed European and North American ice sheets. Although it was long unclear whether there had been a single or multiple glaciations, this controversy ended with the large-scale mapping of two distinct moraine systems in North America (Chamberlin, 1894). In the European Alps, the systematic classification of the glaciofluvial terraces on the northern foreland later indicated that there had been at least four major glaciations in the Alps (Penck and Brückner, 1909). More recent studies of glaciofluvial stratigraphy in the
foreland indicate at least 15 Pleistocene glaciations (Schlüchter, 1988; Ivy-Ochs et al., 2008; Preusser et al., 2011).

From the mid-twentieth century, palaeo-climate records extracted from deep sea sediments and ice cores have provided a much more detailed picture of the Earth's environmental history (e.g., Emiliani, 1955; Shackleton and Opdyke, 1973; Dansgaard et al., 1993; Augustin et al., 2004), indicating several tens of glacial and interglacial periods. During the last 800 ka (thousand years before present), these *glacial cycles* have succeeded each other with a periodicity about 100 ka (Hays et al.,
1976; Augustin et al., 2004). Nevertheless, this global signal is largely governed by the North American and Eurasian ice sheet complexes. It is thus unclear whether glacier advances and retreats in the Alps were in pace with global sea-level fluctuations. Besides, the landform record is typically sparse in time and, most often, spatially incomplete. Palaeo-ice sheets did not leave a continuous imprint on the landscape, and much of this evidence has been overprinted by subsequent glacier re-advances and other geomorphological processes (e.g., Kleman, 1994; Kleman et al., 2006, 2010). Dating uncertainties typically increase
with age, such that dated reconstructions are strongly biased towards later glaciations (Heyman et al., 2011). In the European Alps, although sparse geological traces indicate that the last glacial cycle may have comprised two or three cycles of glacier growth and decay (Preusser, 2004; Ivy-Ochs et al., 2008), most glacial features currently left on the foreland present a record of the last major glaciation of the Alps, dating from the *Last Glacial Maximum* (LGM; Ivy-Ochs, 2015; Wirsig et al., 2016; Monegato et al., 2017).

The spatial extent and thickness of Alpine glaciers during the LGM, an integrated footprint of thousands of years of climate history and glacier dynamics, have been reconstructed from moraines and trimlines across the mountain range (e.g., Penck and Brückner, 1909; Castiglioni, 1940; van Husen, 1987; Bini et al., 2009; Coutterand, 2010). The assumption that trimlines, the upper limit of glacial erosion, represent the maximum ice surface elevation, has been repeatedly invalidated in other glaciated regions of the globe (e.g., Kleman, 1994; Kleman et al., 2010; Fabel et al., 2012; Ballantyne and Stone, 2015). Although no
evidence for cold-based glaciation has been reported in the Alps to date, it is supported by numerical modelling (Blatter and Haeberli, 1984; Haeberli and Schlüchter, 1987; Cohen et al., 2018). Alpine ice flow patterns were primarily controlled by subglacial topography, but there is evidence for flow accross major mountain passes (e.g., Coutterand, 2010; Kelly et al., 2004; van Husen, 2011), and self-sustained ice domes (Bini et al., 2009). Finally, the timing of the LGM in the Alps (Ivy-Ochs et al., 2008; Monegato et al., 2017) is in good agreement with the maximum expansion of continental ice sheets recorded by the
Marine Oxygen Isotope Stage (MIS) 2 (29–14 ka; Lisiecki and Raymo, 2005). Regional variation between different piedmont

lobes exists (Wirsig et al., 2016, Fig. 5), but it is unclear whether this relates to climate or glacier dynamics (Monegato et al., 2017), uncertainties in the dating methods, or both.

Although the glacial history of the European Alps has been studied for nearly three hundred years, uncertainties remain on (1) what climate evolution led to the known maximum ice limits, (2) to what extent ice flow was controlled by subglacial topography, (3) what drove the different responses of the individual lobes, (4) how far above the trimline was the ice surface located, and (5) how many advances occurred during the last glacial cycle.

Here, we intend to explore these open questions from a new angle and use the Parallel Ice Sheet Model (PISM, the PISM authors, 2017), a numerical ice sheet model that approximates glacier sliding and deformation (Sect. 2), to model Alpine glacier dynamics through the last glacial cycle (120–0 ka), a period for which palaeo-temperature proxies are available, albeit non regional. In an attempt to analyse the longstanding questions outlined above, we test the model sensitivity to multiple palaeo-climate forcing (Sect. 3), and then explore the modelled glacier dynamics at high resolution for the optimal forcing (Sect. 4). Additional geological research will be needed to complete our knowledge.

## 2  Ice sheet model set-up

### 2.1  Overview

We use the Parallel Ice Sheet Model (PISM, development version e9d2d1f), an open source, finite difference, shallow ice sheet model (the PISM authors, 2017). The model requires input on initial bedrock and glacier topographies, geothermal heat flux and climate forcing. It computes the evolution of ice extent and thickness over time, the thermal and dynamic states of the ice sheet, and the associated lithospheric response. The model set-up used here was previously developed and tested on the former Cordilleran ice sheet (Seguinot, 2014; Seguinot et al., 2014, 2016) and subsequently adapted for steady climate (Becker et al., 2016) and regional (Jouvet et al., 2017a; Becker et al., 2017) applications to the European Alps.

Ice deformation follows a temperature and water-content dependent creep formulation (Sect. 2.2). Basal sliding follows a pseudo-plastic law where the yield stress accounts for till dilatation under high water pressure (Sect. 2.3). Bedrock topography is deflected under the ice load (Sect. 2.4). Surface mass balance is computed using a positive degree-day (PDD) model (Sect. 2.5). Climate forcing is provided by a monthly climatology from interpolated observational data (WorldClim; Hijmans et al., 2005) and the European Centre for Medium-Range Weather Forecasts Reanalysis Interim (ERA-Interim; Dee et al., 2011), amended with temperature lapse-rate corrections (Sect. 2.6), time-dependent temperature offsets, and in some cases, time-dependent palaeo-precipitation reductions (Sect. 3).

Each simulation starts from assumed present-day ice thickness and equilibrium ice and bedrock temperature at 120 ka, and runs to the present. Our modelling domain of 900 by 600 km encompasses the entire Alpine range (Fig. 1). The simulations were run on two different grids, using horizontal resolutions of 2 and 1 km, respectively. All parameter values are summarized in Table 1.

## 2.2 Ice rheology

Ice sheet dynamics are typically modelled using a combination of internal deformation and basal sliding. PISM is a shallow ice sheet model, which implies that the balance of stresses is approximated based on their predominant components. The Shallow Shelf Approximation (SSA) is combined with the Shallow Ice Approximation (SIA) by adding velocity solutions of the two approximations (Winkelmann et al., 2011, Eqs. 7–9 and 15). In the SIA, topographic roughness is parametrized using a bed smoother range of 5 km (Schoof, 2003). Although the SSA-SIA heuristic is inaccurate in the transition zone where both vertical shear and longitudinal stretch are significant, it was shown to effectively approximate complex ice sheet flow dynamics over mountainous topographies similar to that of the European Alps (Golledge et al., 2012; Ziemen et al., 2016).

Ice deformation is governed by the constitutive law for ice (Glen, 1952; Nye, 1953). Ice softness depends on ice temperature, pressure, and water content, through an enthalpy scheme (Table 1; Aschwanden et al., 2012). It follows a piece-wise Arrhenius-type rheology representative of Holocene polar ice (Table 1; Cuffey and Paterson, 2010, p. 77), and increases with liquid water fractions up to 0.01 (Duval, 1977; Lliboutry and Duval, 1985; Cuffey and Paterson, 2010, p. 65–66), an arbitrary threshold above which new ice deformation measurements are critically needed (Kleiner et al., 2015).

Surface air temperature derived from the climate forcing (Sects. 2.6, 3.1) provides the upper boundary condition to the ice enthalpy model. Temperature is computed in the ice and in the bedrock down to a depth of 3 km below the glacier base, where it is conditioned by a lower boundary geothermal heat flux estimate from multiple geothermal proxies (Goutorbe et al., 2011, similarity method; Fig. 1a).

## 2.3 Basal sliding

A pseudo-plastic sliding law relates the bed-parallel shear stresses to the sliding velocity. The yield stress is modelled using the Mohr–Coulomb criterion (Table 1; Tulaczyk et al., 2000, Eq. 18). using a constant basal friction angle corresponding to the average of available measurements (Cuffey and Paterson, 2010, p. 268). Effective pressure is related to the ice overburden stress and the modelled amount of subglacial water, using a formula derived from laboratory experiments with till extracted from the base of Ice Stream B in West Antarctica (Table 1; Tulaczyk et al., 2000; Bueler and van Pelt, 2015, Eqs 23 and 24). Basal meltwater is accumulated locally without horizontal transport. When the till becomes saturated, additional meltwater assumed to drain instantaneously and is removed from the system in an accountable way. Other parameters (Table 1) follow simulations of the Greenland ice sheet (Aschwanden et al., 2013), or benchmarks when other data is missing (Bueler and van Pelt, 2015).

## 2.4 Basal topography

The initial basal topography is bilinearly interpolated from the hole-filled, Shuttle Radar Topography Mission (SRTM) data with a resolution of 30 arc-sec (Fig. 1b; Jarvis et al., 2008). These data include post-glacial sediment fills and lakes surface topography. However, they were corrected for an estimation of present-day ice thickness based on modern glacier outlines, surface topography and simplified ice physics (Fig. 1c; Huss and Farinotti, 2012).

Basal topography responds to ice load following a bedrock deformation model that includes local isostasy, elastic lithosphere flexure and viscous asthenosphere deformation in an infinite half-space (Lingle and Clark, 1985; Bueler et al., 2007). Model parameters were set according to results from glacial isostatic adjustment modelling of deglacial rebound in the Alps most closely reproducing observed modern uplift rates (Mey et al., 2016, Supplementary Fig. 7).[1]

## 2.5 Surface mass balance

Ice surface accumulation and ablation are computed from monthly mean near-surface air temperature, $T_m$, monthly standard deviation of near-surface air temperature, $\sigma$, and monthly precipitation, $P_m$, using a temperature-index model (e.g., Hock, 2003). Accumulation is equal to precipitation when air temperatures are below $0\,°C$, and decreases to zero linearly with temperatures between 0 and $2\,°C$. Ablation is computed from PDD, the integral of temperatures above $0\,°C$.

The PDD computation accounts for stochastic temperature variations by assuming a normal temperature distribution with standard deviation, $\sigma$, around the expected value, $T_m$. It is expressed by an error-function formulation (Calov and Greve, 2005) which is numerically approximated using week-long sub-intervals. In order to account for the effects of spatial and seasonal variations of temperature variability (Seguinot, 2013), $\sigma$ is computed from ERA-Interim daily temperature values from 1979 to 2012 (Mesinger et al., 2006), including variability associated with the seasonal cycle (Seguinot, 2013), and bilinearly interpolated to the model grids (Fig. 1d and e). Degree-day factors for snow and ice melt are set to values used in the European Ice Sheet Modelling INiTiative (Table 1; EISMINT, Huybrechts, 1998).

## 2.6 Reference climate forcing

The climate forcing driving the ice sheet simulations consists spatially of a present-day monthly mean climatology, $\{T_{m0}, P_{m0}\}$, modified by a temperature lapse-rate correction, $\Delta T_{LR}$, temperature offset time series, $\Delta T_{TS}$, and time-dependent palaeo-precipitation corrections, $\Psi_{PP}$:

$$T_m(t,x,y) = T_{m0}(x,y) + \Delta T_{LR}(t,x,y) + \Delta T_{TS}(t), \tag{1}$$

$$P_m(t,x,y) = P_{m0}(x,y) \cdot \Psi_{PP}(t), \tag{2}$$

The present-day monthly mean climatology was bilinearly interpolated from near-surface air temperature and precipitation rate fields from WorldClim (Hijmans et al., 2005), representative of the period 1960 to 1990. Modern climate of the European Alps is characterised by a north-south gradient in winter and summer air temperatures (Fig. 1f and g), and an east-west gradient in winter precipitation (Fig. 1h) which is reversed in summer (Fig. 1i). WorldClim data were selected as an input to the ice sheet model because they incorporate observations from the dense weather station network of central Europe (Hijmans et al., 2005, Fig. 1). Besides, WorldClim data were previously used as climate forcing for PISM to model the LGM extent of the former

---

[1]Lithosphere rigidity was computed using the erroneous formula, $D = YE^3/12(1-\nu)$, in Mey et al. (2016). The correct formula is $D = YE^3/12(1-\nu^2)$ (Love, 1906, p. 443). Using the Young's modulus, $Y = 100\,GPa$, and the Poisson ratio, $\nu = 0.25$ (Mey et al., 2016), the consequence of this error is that the simulations effectively use an elastic thickness of the lithosphere of $E = 53.8\,km$ instead of $50\,km$ which is well within uncertainties (Mey et al., 2016), and thus introduces a $\sqrt[4]{1+\nu} - 1 = 5.7\,\%$ change in the lenght scale of bedrock deformation (Walcott, 1970).

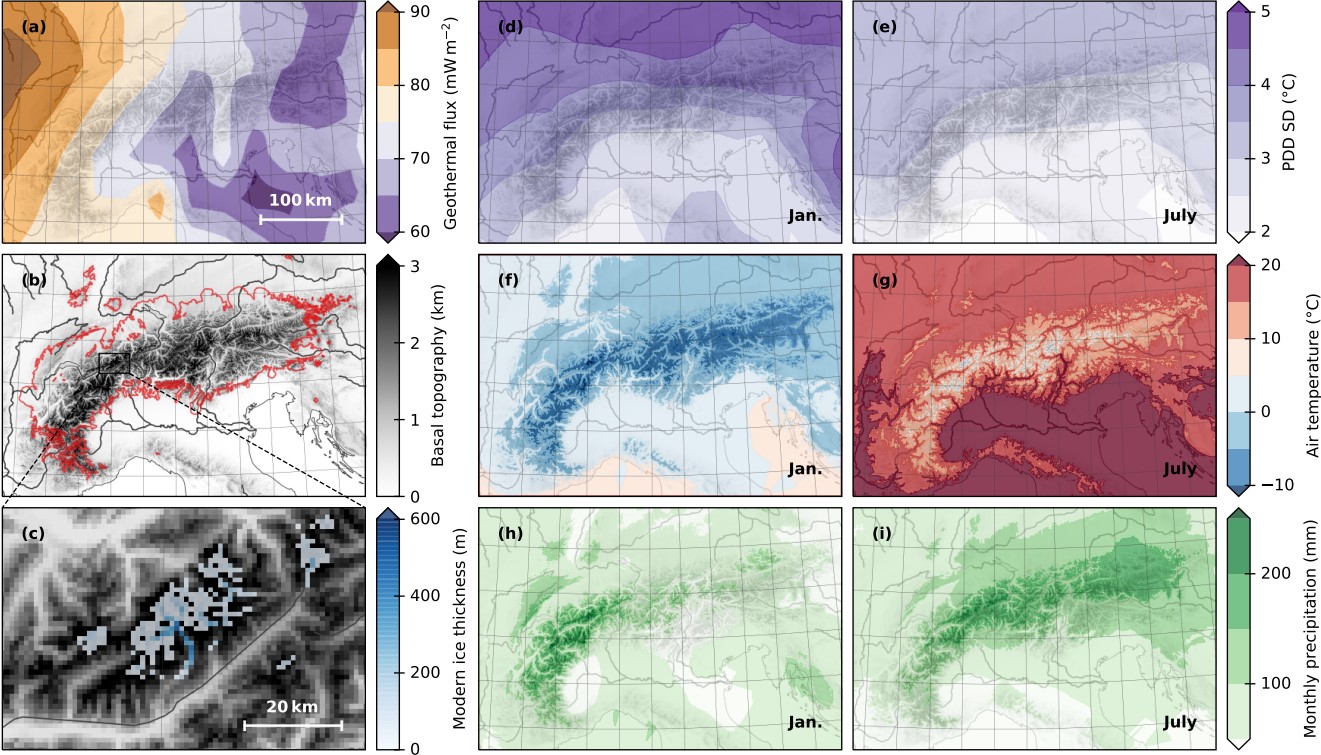

**Figure 1. (a)** Geothermal heat flow from applying the similarity method to multiple geophysical proxies (Goutorbe et al., 2011) used as a boundary condition to the bedrock thermal model 3 km below the ice-bedrock interface. **(b)** Initial basal topography from SRTM (Jarvis et al., 2008) and geomorphological reconstruction of Last Glacial Maximum Alpine glacier extent (solid red line, Ehlers et al., 2011). **(c)** Extract from the estimated present-day ice thickness (Huss and Farinotti, 2012) substracted from the SRTM topography and aggregated to a 1 km horizontal resolution. **(d)** Modern January and **(e)** July standard deviation (Seguinot, 2013) of daily mean temperature from the ERA-Interim (1979–2012; Dee et al., 2011) monthly climatology approximating temperature variability. **(f)** Modern January and **(g)** July mean near-surface air temperature, and **(h)** January and **(i)** July precipitation from WorldClim (1960–1990; Hijmans et al., 2005) used to compute surface mass balance. The background maps contain Natural Earth Data (Patterson and Kelso, 2017).

Cordilleran ice sheet in good agreement with geological evidence along the southern margin (Seguinot et al., 2014) where weather station density is lower than in the Alps. Finally, the last glacial cycle Alpine glaciers did not extend over marine areas where WorldClim data are missing.

The temperature lapse-rate corrections, $\Delta T_{\mathrm{LR}}$, are computed as a function of ice surface elevation, $s$, using the SRTM-based topography data provided with WorldClim as a reference, $b_{\mathrm{ref}}$:

$$\Delta T_{\mathrm{LR}}(t,x,y) = -\gamma[s(t,x,y) - b_{\mathrm{ref}}] \tag{3}$$

$$= -\gamma[h(t,x,y) + b(t,x,y) - b_{\mathrm{ref}}], \tag{4}$$

thus accounting for the evolution of ice thickness, $h = s - b$, on the one hand, and for differences between the basal topography of the ice flow model, $b$, and the WorldClim reference topography, $b_{\mathrm{ref}}$, on the other hand. All simulations use an annual temperature lapse rate of $\gamma = 6\,\mathrm{K\,km^{-1}}$, which is slightly above annual lapse rates measured in the Alps but more representative of summer months when surface melt occurs (Rolland, 2003, Fig. 3).

## 3 Palaeo-climate forcing

In this section, we analyze the model sensitivity to palaeo-climate forcing through the last glacial cycle, using three palaeo-temperature records (Sect. 3.1) and two parametrizations of palaeo-precipitation (Sect. 3.2), in terms of modelled evolution of total ice volume (Sect. 3.3) and glaciated area during MIS 2 and 4 (Sect. 3.4).

These simulations use a horizontal resolution of 2 km. The vertical grid consists of 31 temperature layers in the bedrock and up to 126 enthalpy layers in the ice, corresponding to vertical resolutions of 100 and 40 m, respectively.

### 3.1 Palaeo-temperature forcing

Only few regional proxy records exist that extend over periods when the Alps were glaciated (Heiri et al., 2014). These include lake sediment records in north and west of the Alps (de Beaulieu and Reille, 1992; Wohlfarth et al., 2008; Duprat-Oualid et al., 2017, e.g.,), and cave speleothems in the Eastern (e.g., Spötl and Mangini, 2002; Boch et al., 2011) and Western (Luetscher et al., 2015) Alps. Due to the scarcity of vegetation north of the Alps during glacial periods, varying sources for moisture advection, and the limited duration of the records, quantitative palaeoclimatic interpretation will require combining multiple proxies in space and time, and comparing them against regional circulation model output (Heiri et al., 2014).

In our simulations, temperature offset time-series, $\Delta T_{\mathrm{TS}}$, are derived from distal palaeo-temperature proxy records from the Greenland Ice Core Project (GRIP; Dansgaard et al., 1993), the European Project for Ice Coring in Antarctica (EPICA; Jouzel et al., 2007), and an oceanic sediment core from the Iberian margin (MD01-2444; Martrat et al., 2007). Palaeo-temperature anomalies from the GRIP record are calculated from the oxygen isotope ($\delta^{18}\mathrm{O}$) measurements using a quadratic equation (Johnsen et al., 1995),

$$
\begin{aligned}
\Delta T_{\mathrm{TS}}(t) = {} & -11.88[\delta^{18}\mathrm{O}(t) - \delta^{18}\mathrm{O}(0)] \\
& -0.1925[\delta^{18}\mathrm{O}(t)^2 - \delta^{18}\mathrm{O}(0)^2],
\end{aligned}
\tag{5}
$$

while temperature reconstructions from the EPICA and MD01-2444 records are provided as such. For each proxy record used and each of the parameter set-ups used in the sensitivity tests, palaeo-temperature anomalies are scaled linearly (Table 2, Fig. 2a) so that, within a $150 \times 100\,\mathrm{km}$ rectangular domain covering the Rhine glacier piedmont lobe (Fig. 3a, black rectangle), the modelled cumulative glaciated area during MIS 2 (29–14 ka) is consistent with the glaciated area of the geological reconstruction (Ehlers et al., 2011).

**Table 1.** Parameter values used in the ice sheet model. Symbols refer to equations used by Seguinot (2014); Seguinot et al. (2016).

| Not. | Name | Value | Unit | Source |
|------|------|-------|------|--------|
| **Ice rheology** | | | | |
| $\rho$ | Ice density | 910 | $\mathrm{kg\,m^{-3}}$ | Aschwanden et al. (2012) |
| $g$ | Standard acceleration due to gravity | 9.81 | $\mathrm{m\,s^{-2}}$ | Aschwanden et al. (2012) |
| $n$ | Glen exponent | 3 | – | Cuffey and Paterson (2010, p. 55–57) |
| $A_c$ | Ice hardness coefficient cold | $2.847 \times 10^{-13}$ | $\mathrm{Pa^{-3}\,s^{-1}}$ | Cuffey and Paterson (2010, p. 72) |
| $A_w$ | Ice hardness coefficient warm | $2.356 \times 10^{-2}$ | $\mathrm{Pa^{-3}\,s^{-1}}$ | Cuffey and Paterson (2010, p. 72) |
| $Q_c$ | Flow law activation energy cold | $6.0 \times 10^4$ | $\mathrm{J\,mol^{-1}}$ | Cuffey and Paterson (2010, p. 72) |
| $Q_w$ | Flow law activation energy warm | $11.5 \times 10^4$ | $\mathrm{J\,mol^{-1}}$ | Cuffey and Paterson (2010, p. 72) |
| $E_{\mathrm{SIA}}$ | SIA enhancement factor | 2 | – | Cuffey and Paterson (2010, p. 77) |
| $E_{\mathrm{SSA}}$ | SSA enhancement factor | 1 | – | Cuffey and Paterson (2010, p. 77) |
| $T_c$ | Flow law critical temperature | 263.15 | K | Paterson and Budd (1982) |
| $f$ | Flow law water fraction coeff. | 181.25 | – | Lliboutry and Duval (1985) |
| $R$ | Ideal gas constant | 8.31441 | $\mathrm{J\,mol^{-1}\,K^{-1}}$ | Cuffey and Paterson (2010, p. 72) |
| $\beta$ | Clapeyron constant | $7.9 \times 10^{-8}$ | $\mathrm{K\,Pa^{-1}}$ | Lüthi et al. (2002) |
| $c_i$ | Ice specific heat capacity | 2009 | $\mathrm{J\,kg^{-1}\,K^{-1}}$ | Aschwanden et al. (2012) |
| $c_w$ | Water specific heat capacity | 4170 | $\mathrm{J\,kg^{-1}\,K^{-1}}$ | Aschwanden et al. (2012) |
| $k$ | Ice thermal conductivity | 2.10 | $\mathrm{J\,m^{-1}\,K^{-1}\,s^{-1}}$ | Aschwanden et al. (2012) |
| $L$ | Water latent heat of fusion | $3.34 \times 10^5$ | $\mathrm{J\,kg^{-1}\,K^{-1}}$ | Aschwanden et al. (2012) |
| **Basal sliding** | | | | |
| $q$ | Pseudo-plastic sliding exponent | 0.25 | – | Aschwanden et al. (2013) |
| $v_{\mathrm{th}}$ | Pseudo-plastic threshold velocity | 100 | $\mathrm{m\,a^{-1}}$ | Aschwanden et al. (2013) |
| $c_0$ | Till cohesion | 0 | Pa | Tulaczyk et al. (2000) |
| $e_0$ | Till reference void ratio | 0.69 | – | Tulaczyk et al. (2000) |
| $C_c$ | Till compressibility coefficient | 0.12 | – | Tulaczyk et al. (2000) |
| $\delta$ | Minimum effective pressure ratio | 0.02 | – | Bueler and van Pelt (2015) |
| $\phi$ | Till friction angle | 30 | ° | Cuffey and Paterson (2010, p. 268) |
| $W_{\mathrm{max}}$ | Maximum till water thickness | 2 | m | Bueler and van Pelt (2015) |
| **Bedrock and lithosphere** | | | | |
| $\rho_b$ | Bedrock density | 3300 | $\mathrm{kg\,m^{-3}}$ | – |
| $c_b$ | Bedrock specific heat capacity | 1000 | $\mathrm{J\,kg^{-1}\,K^{-1}}$ | – |
| $k_b$ | Bedrock thermal conductivity | 3 | $\mathrm{J\,m^{-1}\,K^{-1}\,s^{-1}}$ | – |
| $\nu_m$ | Astenosphere viscosity | $2.2 \times 10^{20}$ | $\mathrm{Pa\,s}$ | Mey et al. (2016) |
| $\rho_m$ | Astenosphere density | 3300 | $\mathrm{kg\,m^{-3}}$ | Mey et al. (2016) |
| $D$ | Lithosphere flexural rigidity | $1.389 \times 10^{24}$ | $\mathrm{N\,m}$ | Mey et al. (2016) |
| **Surface and atmosphere** | | | | |
| $T_s$ | Temperature of snow precipitation | 273.15 | K | – |
| $T$ | Temperature of rain precipitation | 275.15 | K | |

**Table 2.** Palaeo-temperature proxy records and scaling factors yielding temperature offset time-series used to force the ice sheet model through the last glacial cycle (Fig. 2). $f$ corresponds to the scaling factor adopted to yield Last Glacial Maximum ice limits in the vicinity of mapped end moraines (Fig. 3a), and $[\Delta T_{TS}]_{32}^{22}$ refers to the resulting mean temperature anomaly during the period 32 to 22 ka after scaling.

| Forcing | Latitude | Longitude | Elev. (m a.s.l.) | Proxy | $f$ | $[\Delta TS]_{32}^{22}$ (K) | Reference |
|---|---|---|---|---|---|---|---|
| GRIP | | | | | 0.50 | −8.2 | |
| GRIP, PP | 72°35′ N | 37°38′ W | 3238 | $\delta^{18}O$ | 0.63 | −10.4 | Dansgaard et al. (1993) |
| EPICA | | | | | 1.05 | −9.7 | |
| EPICA, PP | 75°06′ S | 123°21′ E | 3233 | $\delta^{18}O$ | 1.33 | −12.2 | Jouzel et al. (2007) |
| MD01-2444 | | | | | 1.84 | −8.0 | |
| MD01-2444, PP | 37°34′ N | 10°04′ W | −2637 | $U_{37}^{K'}$ | 2.44 | −10.6 | Martrat et al. (2007) |

## 3.2 Palaeo-precipitation forcing

Finally, in some simulations (hereafter labelled PP), precipitation was reduced with air temperature in order to simulate the potential rarefaction of atmospheric moisture in colder climates. This was done using an empirical relationship derived from observed accumulation rates and oxygen isotopes concentrations in the GRIP ice core (Dahl-Jensen et al., 1993),

$$\Psi_{PP}(t) = \exp[\psi \Delta T_{TS}(t)], \tag{6}$$

with $\psi = 0.169/2.4 = 0.0704$ (Huybrechts, 2002). The control simulations use constant precipitation, corresponding to $\psi = 0$. This simple relationship does not reflect the complexity of atmospheric circulation changes that governed moisture availability over the Alps during the last glacial cycle. Palaeoclimate proxies indicate slightly reduced LGM precipitation in western Europe with anomalies diminishing eastwards (Wu et al., 2007). Regional circulation models indicate generally dryer conditions during MIS 3 (Barron and Pollard, 2002; Kjellström et al., 2010) but more precipitation south of the Alps during MIS 2 (Strandberg et al., 2011; Ludwig et al., 2016). Besides, changes in the ice surface topography may have redistributed orographic precipitation.

## 3.3 Sensitivity of ice volume evolution

For the three palaeo-temperature records and the two palaeo-precipitation parametrizations used, the model yields significant ice volume build-up during MIS 4 and 2 (Fig. 2b), corresponding to documented glaciation periods in the Alps (Preusser, 2004; Ivy-Ochs et al., 2008). All simulations also yield important glaciations during MIS 5 and 3, but their timing and amplitude varies significantly depending on the climate forcing used. All six simulations overestimate the late-glacial re-advance during the Younger Dryas (12.9–11.7 ka; cf. e.g., Ivy-Ochs et al., 2009). This is partly because the 2 km resolution used in these simulations is too coarse to resolve Younger Dryas Alpine glaciers. However, the large total ice volume overestimation resulting from the GRIP temperature forcing (Fig. 2b, blue curves) certainly indicates that the vigorous cooling experienced during the Younger Dryas in Greenland is not representative of the Alps.

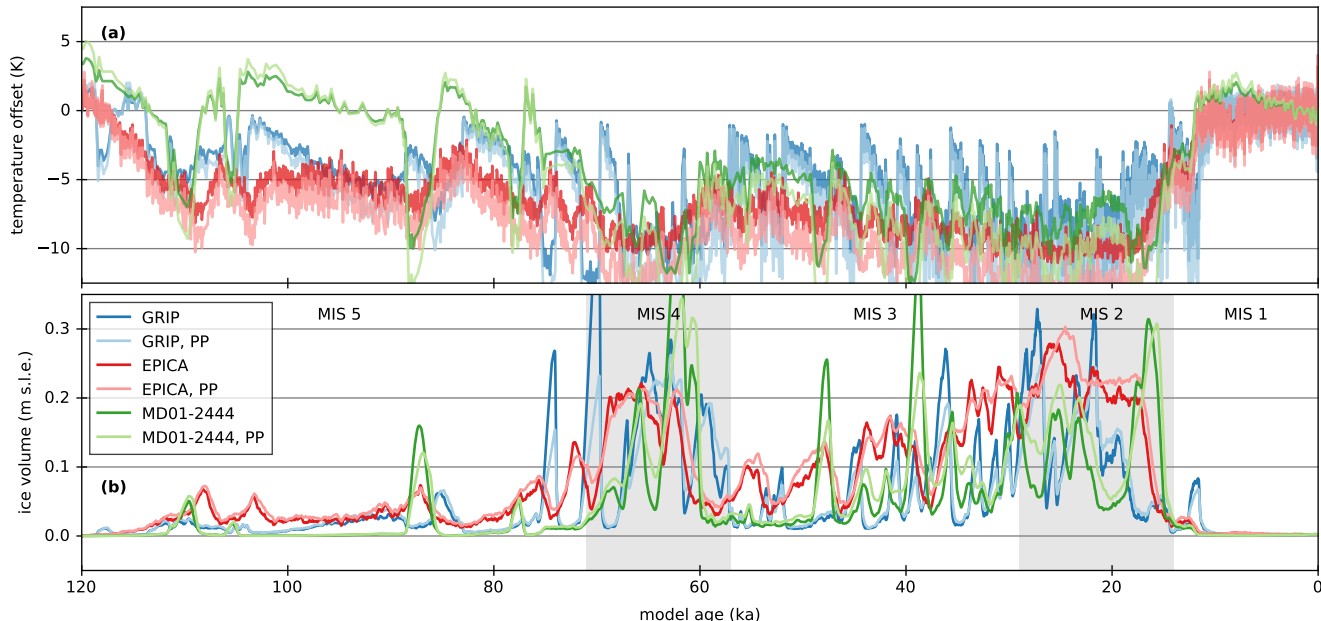

**Figure 2. (a)** Temperature offset time-series from ice core and ocean records (Table 2) used as palaeo-climate forcing for the ice sheet model. **(b)** Modelled total ice volume through the last 120 thousand years (ka) expressed in meters of sea-level equivalent (m s.l.e.). Shaded gray areas indicate the timing for MIS 2 and MIS 4 (Lisiecki and Raymo, 2005). The simulation driven by the EPICA temperature yields smaller ice volume variability and a more realistic timing of deglaciation.

Geological data from the best documented Alpine piedmont lobes indicate that their maximum extension during MIS 2 occurred at around 25.5–24.0 ka (Monegato et al., 2017), after which Alpine glaciers remained or potentially re-advanced to within close reach of this maximum extent, until as late as 22 to 17 ka (Ivy-Ochs, 2015; Wirsig et al., 2016, Fig. 5). The EPICA simulations yield an early maximum ice volume at 25.2/24.6 (without/with palaeo-precipitation reductions) followed

by a retreat and then a standstill until 17.3 ka (Fig. 2b, red curves) in a very good agreement with these geological data. On the other hand, the simulations forced by the GRIP palaeo-temperature record yield two distinct total ice volume maxima at 27.3/27.0 and 21.8/21.7 ka followed by an early deglaciation of the foreland by 21.4 ka. The MD01-2444 palaeo-temperature forcing yields a late LGM peak ice volume at 16.5/15.7 ka followed by a rapid retreat, in contrast to dated records of large-scale ice retreat by 17 to 16 ka (Ivy-Ochs et al., 2006, 2008).

Finally, all simulations yield very strong ice volume variability over the last 120 ka (Fig. 2b), indicative of many more than the two or three documented cycles of glacier advance and retreats onto the foreland for this time period (Preusser, 2004; Ivy-Ochs et al., 2008). Palaeo-precipitation reductions dampen some of the small-scale variability, but they have little effect on the millennial scales that characterise those cycles (Fig. 2b, light colour curves). In particular, strong millennial variability recorded in GRIP $\delta^{18}$O (Dansgaard-Oeschger events) have large repercussions on the modelled Alpine ice volume, including around six

glaciations of LGM magnitude (Fig. 2b, blue curves), which is not supported by geologic evidence (Preusser, 2004; Ivy-Ochs

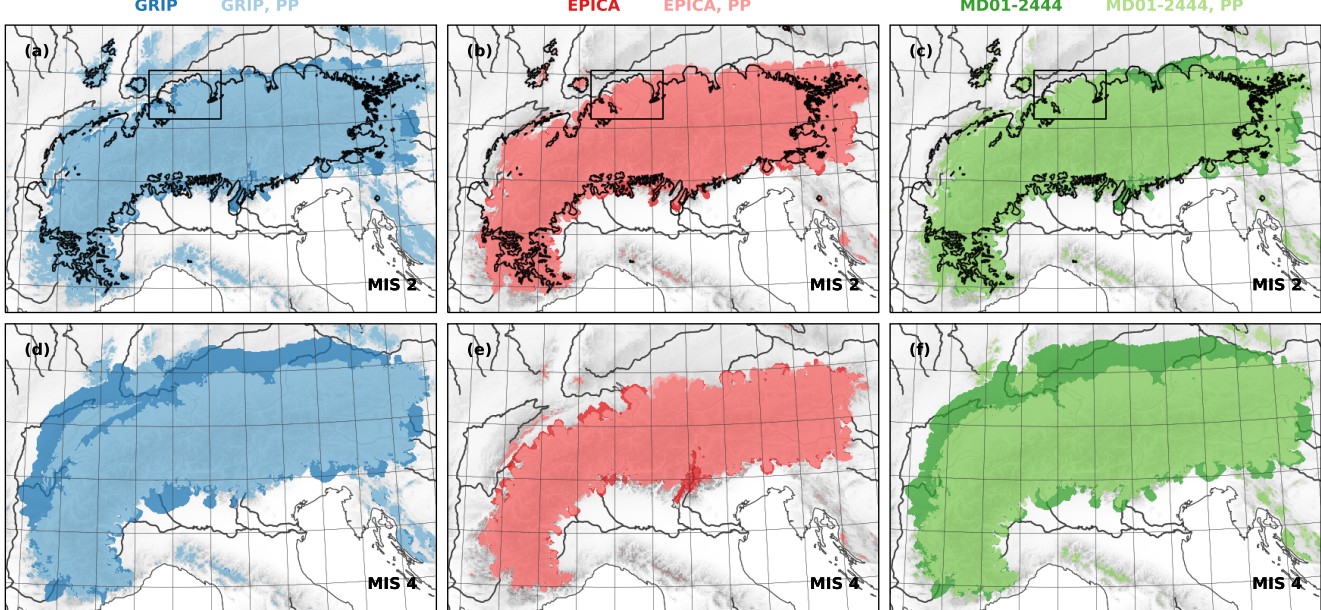

**Figure 3. (a–c)** Modelled maximum ice extent during MIS 2 (29–14 ka), without (dark colours) and with (light colours) palaeo-precipitation corrections, and using temperature time-series scaling factors (Table 2) adjusted to model the area glaciated by the Rhine glacier piedmont lobe (black rectangle) in agreement with the Last Glacial Maximum (LGM) geomorphological reconstruction (solid black line, Ehlers et al., 2011). **(d–f)** Modelled maximum ice extent during MIS 4 (71–57 ka). Only the simulation driven by the EPICA temperature time-series yields realistic MIS 4 ice cover.

et al., 2008). Dansgaard-Oeschger events, however, have been recorded in European lake sediments (Wohlfarth et al., 2008) and cave speleothems (Spötl and Mangini, 2002; Luetscher et al., 2015). Our results indicate that their magnitude in Europe was likely smaller than that recorded in GRIP $\delta^{18}$O, which either implies smaller associated palaeo-temperature fluctuations in Europe than Greenland, or temperature-independent controls on GRIP $\delta^{18}$O. In contrast, the EPICA palaeo-temperature record has the smallest variability among the records used (Fig. 2a, red curves). This results in smaller variability in the total ice volume, and more restrictive glaciations during MIS 5 and 3, (Fig. 2b, red curves).

### 3.4 Sensitivity of glaciated area

During MIS 2, all six simulations yield comparative modelled maximum extents (Fig. 3a). This result is inherent to the palaeo-climate forcing approach used, which involves a linear scaling of each palaeo-temperature anomaly record to a geomorphological reconstruction of the area glaciated by the Rhine Glacier piedmont lobe (Sect. 3.1; Fig. 3a, black rectangle). Outside this benchmark area, all simulations underestimate ice extent in the western part of the model domain, and overestimate it in the eastern part, relative to the geomorphological ice limits (Fig. 3a). This result might indicate that the LGM temperature depression, here taken as homogeneous, was actually lower in the Eastern Alps than in the Western Alps, as previously shown

by positive degree-day modelling of the central European palaeo-ice caps (Heyman et al., 2013). Alternatively, the east-west gradient in winter precipitation existing today (Fig. 1h) was perhaps enhanced during the LGM, as indicated by pollen reconstructions (Wu et al., 2007). The LGM extent of Alpine glaciers might have been affected by an east-west gradient in variables not accounted for by the PDD model, such as cloud cover or dust deposition. Finally, modern precipitation data from WorldClim also bear uncertainties and exhibit local disagreement with other regional data (Isotta et al., 2013).

Besides this general pattern, there exist differences in glaciated area depending on the palaeo-climate forcing used. The MD01-2444, and to a greater extent, the GRIP palaeo-temperature records, tend to overestimate MIS 2 ice cover on all peripheral ranges that surround the Alps (Fig. 3a and c). This is particularly true when no precipitation corrections are applied (bright colours). In fact, both palaeo-temperature records have a larger temperature variability and contain brief spells of cold temperatures (Fig. 2b, blue and green curves), which are too short to develop a fully grown Alpine ice sheet, but long enough to build up ice cover on a smaller scale on these peripheral ranges. On the other hand, peripheral glaciation modelled using the EPICA record (Fig. 3b), which has smaller temperature variability, is in relatively good agreement with the geomorphological reconstructions.

The modelled extent of glaciation during MIS 4 depicts a more pronounced sensitivity to the choice of palaeo-climate forcing used. Using the GRIP and MD01-2444 palaeo-temperature records, Alpine glaciers are modelled to extend well beyond the reach of documented LGM end-moraines (Fig. 3d and f). Because, in both cases, the modelled glaciation extent corresponds to brief cold spells in the palaeo-temperature forcing (Fig. 2a), palaeo-precipitation reductions greatly reduce the excessive modelled ice cover (Figs. 2b and 3d and f, light colours). However, with or without palaeo-precipitation corrections, the GRIP and MD01-2444 forcing yield modelled ice extents (Fig. 3d and f) and volumes (Fig. 2b) considerably larger during MIS 4 than during MIS 2, which is not supported by geological evidence (Preusser, 2004; Ivy-Ochs et al., 2008; Husen and Reitner, 2011; Barrett et al., 2017; Haldimann et al., 2017). Gravel deposits from the Rhine Glacier indicate an early last glacial cycle foreland glaciation, but the extent reached is less than that in MIS 2 (Keller and Krayss, 2010). Dropstones in lake sediments indicate a late MIS 4 advance restricted to the upper Inn Valley (Barrett et al., 2017). For the Linth Glacier, no evidence for a pre-LGM glacier advance during the last glacial cycle is available (Haldimann et al., 2017).

The EPICA palaeo-temperature forcing, on the other hand, yields a MIS 4 glaciation that is only slightly less expansive than the MIS 2 glaciation (Fig. 3e). The sensitivity of the glaciated area to palaeo-precipitation reductions is mostly limited to the southern terminal lobes of central-Alpine glaciers, where reduced precipitation results in a slightly lesser extent during both MIS 2 and 4 (Fig. 3b and e).

Based on the above considerations on timing of the LGM during MIS 2, and modelled ice extent during MIS 4, we select EPICA as our optimal palaeo-temperature record for a more detailed and higher-resolution comparison of modelled glacier dynamics to available geological evidence (Sect. 4). As a conservative approach in regard to the rapid ice volume fluctuations, we choose to include palaeo-precipitation corrections in the following higher-resolution run.

## 4   Results and discussion

In this section, we compare the model output to geological evidence from the last glacial cycle, in terms of Last Glacial Maximum extent (Sect. 4.1), ice flow patterns (Sect. 4.2), timing of the Last Glacial Maximum (Sect. 4.3), ice thickness (Sect. 4.4), and glacial cycle dynamics (Sect. 4.5).

This simulation is forced by the optimal EPICA palaeo-temperature record (Sect. 3.1) and includes palaeo-precipitation reductions (Sect. 3.2). It uses a horizontal resolution of 1 km. The vertical grid consists of 61 temperature layers in the bedrock and up to 251 enthalpy layers in the ice, corresponding to vertical resolutions of 50 and 20 m, respectively.

### 4.1   Last Glacial Maximum ice extent

The LGM extent of Alpine glaciers has been mapped with varying level of detail across the Western (Jäckli, 1962; Bini et al., 2009; Coutterand, 2010; Buoncristiani and Campy, 2011; Hantke, 2011), and Eastern Alps (Penck and Brückner, 1909; Castiglioni, 1940; van Husen, 1987; BGR, 2007; van Husen, 2011; Bavec and Verbič, 2011). Some of these maps have been compiled into a reconstruction, covering the entire Alps (Ehlers et al., 2011), and reproduced here (Fig. 4, red line) for comparison against model results.

The modelled total ice volume reaches a maximum of $123 \times 10^3 \text{ km}^3$, or 300 mm of sea-level equivalent, at 24.56 ka (Fig. 4b). A maximum glacierized area of $163 \times 10^3 \text{ km}^2$ is attained shortly afterwards at 24.57 ka (Fig. 4a). Although the modelled timing of the LGM varies accross the mountain range (Sect. 4.3), at 24.57 ka nearly all outlet glaciers extend to within a few kilometres from their modelled maximum stage (Fig. 4a, dashed orange line).

The palaeo-climate forcing was adapted (Sect. 3.1) to model the maximum configuration of the Rhine Glacier in broad agreement with the geological reconstruction of the LGM extent (Fig. 4a, solid red line), yet discrepancies remain elsewhere. As already outlined for low resolution runs (Sect. 3.4), the extent of glaciation in the north-western Alps, including the Rhone Glacier complex, the Jura ice cap, and the Lyon Lobe, is underestimated in the model results (Fig. 4a, cf Bini et al., 2009; Coutterand, 2010). On the other hand, the model yields excessive ice cover in the Eastern Alps, where the Mur, Drava and Sava Glaciers are modelled to extend tens of km beyond the mapped ice limits (Fig. 4a, cf van Husen, 1987; Bavec and Verbič, 2011). As previously discussed, these discrepancies might indicate that the LGM climate was characterized by an east-west gradient in temperature (cf. Heyman et al., 2013) or precipitation (cf. Wu et al., 2007) anomalies relative to present, or an east-west gradient in variables that are not accounted for by the PDD model.

On the other hand, while atmospheric circulation models (Strandberg et al., 2011; Ludwig et al., 2016) and palaeo-climate proxies (Luetscher et al., 2015) both support differential precipitation changes north and south of the Alps, our model results show no obvious north-south bias as compared to the mapped LGM margins (Fig. 4a) despite the homogeneous temperature and precipitation anomalies applied relative to present. Although this may appear as a contradiction with previous, constant-climate modelling results (Becker et al., 2016; Jouvet et al., 2017b), it follows from introducing a time-dependent palaeo-temperature forcing. Without introducing differential precipitation change north and south of the Alps, constant climate forcing systematically resulted in extraneous (Becker et al., 2016, Fig. 3) and premature (Becker et al., 2016, Fig. 4) glaciation on

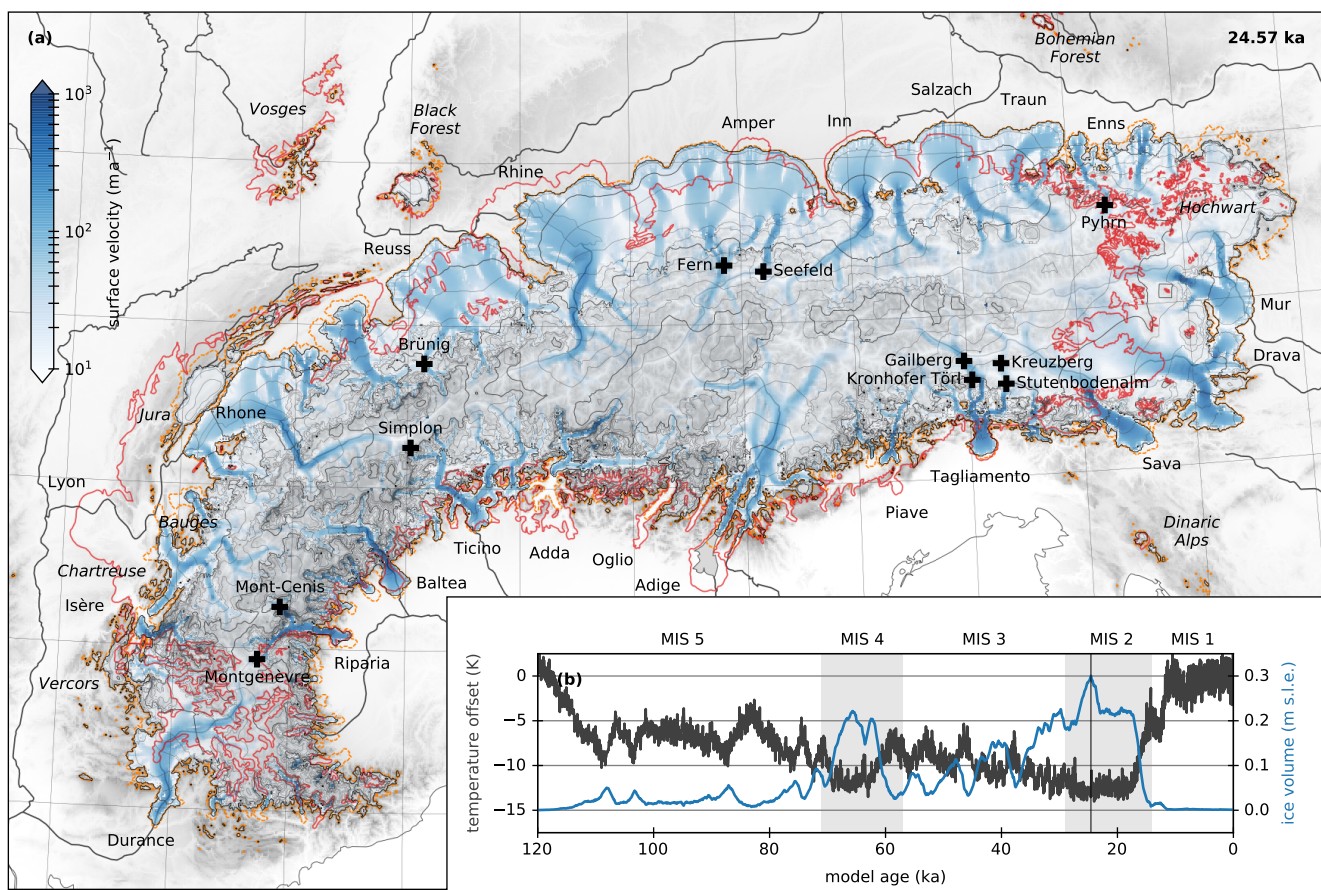

**Figure 4. (a)** Modelled bedrock topography (grey), ice surface topography (200 m contours), and ice surface velocity (blue) in the Alps 24.57 ka before present, corresponding to the modelled maximum ice cover. The modelled LGM ice extent (dashed orange line) and geomorphological reconstruction (solid red line, Ehlers et al., 2011), and some major transfluences across mountain passes (crosses) are shown. **(b)** Temperature offset time-series from the EPICA ice core used as palaeo-climate forcing for the ice flow model (black curve, Jouzel et al., 2007), and modelled total ice volume through the last glacial cycle (120–0 ka), expressed in meters of sea level equivalent (m s.l.e., blue curve). Shaded gray areas indicate the timing for Marine Oxygen Isotope Stage (MIS) 2 and MIS 4 according to a global compilation of benthic $\delta^{18}$O records (Lisiecki and Raymo, 2005). The black vertical line indicates the modelled age of maximum ice cover at 24.57 ka.

the north slope of the Alps. Using time-dependent palaeo-temperature forcing, progressive atmospheric cooling over several thousand years allow for warmer temperatures, closer to the thermodynamic equilibrium, within the ice and the bedrock. This results in thinner glaciers and limits overshoot of the equilibrium state (discussed further in Sect. 4.3).

On a more regional level, the maximum extent of the Adda, Oglio, Adige and Piave Glaciers in the central southern Alps is modelled several kilometres within the mapped LGM moraines (Fig. 4a). This appears to result from the palaeo-precipitation reduction used to force the model (Fig. 3b), which is likely unrealistic in at least this part of the model domain. On the

other hand, the Durance glacier in the south-western Alps is modelled to extend several kilometers beyond the mapped limit (Fig. 4a), indicating that the LGM temperature depression was likely dampened by Mediterranean climate in this part of the model domain. However, geochronological data from the south-western Alps are sparse, and the LGM extent compilation (Ehlers and Gibbard, 2004) is inconsistent with more recent regional reconstructions (Federici et al., 2017).

The model reproduces peripheral ice caps where documented by geological evidence on the Vercors, Chartreuse and Bauge Prealpine reliefs, the Jura Mountains, the Vosges Mountains, the Black Forest, the Bohemian Forest and the Dinaric Alps (Fig. 4a). An independent ice cap also covers the Hochwart massif during most of the simulation, yet it is engulfed by Alpine glaciers during the LGM to become a peripheral ice dome (Fig. 4a; cf. van Husen, 2011, Fig. 2.5). The LGM extent of peripheral ice caps is underestimated in the Vosges and Jura mountains, but it is overestimated for the more meridional Vercors

Massif (Fig. 4a; cf. Coutterand, 2010, Figs. 4.28, 4.32, and 4.33, p. 322–321). Thus, there exists a regional conformity between the model-data discrepancies obtained for the main Alpine ice sheet and those obtained for peripheral ice caps, including too extensive modelled ice cover to the East and extreme South-West, and too restrictive modelled ice cover to the North-West. Because peripheral ice caps have very different glacier dynamics than the main ice sheet, this conformity most likely indicates a climatic cause, rather than an ice dynamics cause, for these discrepancies.

**4.2  Ice flow patterns**

The LGM Alpine ice flow pattern is traditionally described as that of a network of interconnected valley glaciers, primarily controlled by subglacial topography. However, geomorphology shows that ice was thick enough to flow accross high mountain passes throughout the mountain range (e.g., Onde, 1938; Penck and Brückner, 1909; Jäckli, 1962; van Husen, 1985, 2011; Kelly et al., 2004; Coutterand, 2010), and even perhaps to form self-sustained ice domes (Florineth, 1998; Florineth and Schlüchter,

1998; Kelly et al., 2004; Bini et al., 2009).

The modelled flow pattern at 24.57 ka is complex (Fig. 4a, blue colour mapping). Ice velocities of several hundred metres per year, characteristic for basal sliding, generally occur along the main river valleys, while ice domes and ice divides are predominantly located over major reliefs areas, where ice moves only by a few metres per year, characteristic for internal deformation without basal sliding (Fig. 4a). Nevertheless, the model results depict exceptions to this general pattern as occasional ice flow

across the modern water divides, i.e. transfluences.

In the Western Alps, major transfluences occur for instance across Col de Montgenèvre, Col du Mont-Cenis, Simplon Pass and Brünig Pass (Fig. 4a). Although a transfluence across Col de Montgenèvre was previously questioned (Cossart et al., 2012, Fig. 2), evidence for southerly ice flow across Col du Mont-Cenis has been recognized (Onde, 1938; Coutterand, 2010, Fig. 3.18, p. 284). Similarly, tranfluences have been previously identified from the geomorphology across Simplon Pass (Kelly

et al., 2004) and from the distribution of glacial erratics across Brünig Pass (Jäckli, 1962). On the other hand, the simplified climate forcing used in this simulation could not reproduce the transport of glacial erratics from southern Valais to observed deposition sites in the Solothurn region (cf. Jouvet et al., 2017a).

In the Eastern Alps, major transfluences are modelled to have occurred for instance across Fern Pass, the Seefeld Saddle, the Gailberg Saddle, the Kreuzberg Saddle, Kronhof Pass, Studenbodenalm, and Pyhrn Pass (Fig. 4a). Transfluences across

Fern Pass and the Seefeld Saddle are known from the geomorphology (Penck and Brückner, 1909; van Husen, 2011, Fig. 2.4). It is also known that ice flowed across the Gailberg and Kreuzberg saddles (van Husen, 1985), and Pyhrn Pass (van Husen, 2011, Fig. 2.5). On the other hand, no transfluence has previously been documented over the Kronhof Pass, Studenbodenalm or elsewhere over the Karnic Alps. The Tagliamento catchment is usually assumed to not have received tranfluences from the Drava catchment (Monegato et al., 2007), but this would not be incompatible with reconstructed ice surface elevations in this area (van Husen, 1987).

Except for too extensive ice cover in the easternmost part of the model domain (Sect. 4.1), there is generally a good agreement between transfluences observed in the geomorphology and the model results. Both depict the LGM Alpine ice flow pattern as an intermediate between that of a topography-controlled ice field, and that of a self-sustained ice cap.

## 4.3 Timing of the Last Glacial Maximum

The timing of the LGM has been documented by radiocarbon and cosmogenic isotope dating techniques at multiple locations around the Alps (cf. reviews by Ivy-Ochs, 2015; Wirsig et al., 2016; and the more recent publications by Federici et al., 2017; Monegato et al., 2017; Gild et al., 2018; Ivy-Ochs et al., 2018). These data indicate that Alpine glaciers reached their maximum extent between 26 and 20 ka (calibrated $^{14}$C and $^{10}$Be ages), but also that terminal lobes stayed in the foreland until 22 ka to 17 ka, when they experienced rapid retreat synchronously to lowering of the ice surface in the mountains (Ivy-Ochs, 2015; Wirsig et al., 2016, Fig. 5; Monegato et al., 2017, Fig. 3).

In our simulation, the maximum areal cover is reached at 24.57 ka (Fig. 4a), yet individual glacier lobes reach their maximum extent at different ages (Fig. 5). For instance, the Dora Riparia, Dora Baltea, and Tagliamento Glaciers reach an early maximum before 27 ka; the Durance, Rhone, Inn, Enns, Ticino, and Adige Glaciers reach their maximum thickness in phase with the overall Alpine areal maximum around 25 ka; while the Isère, Adda and Oglio Glaciers reach a late maximum after 24 ka (Fig. 5). Remarkably, peripheral ice caps on the Vercors, Jura Mountains, Black Forest and Hochschwab reach their maximum extent even later and locally after 21 ka (Fig. 5).

These differences in timing of the LGM occur in the model results despite the homogeneous temperature and precipitation anomalies supplied as palaeo-climate forcing. The LGM timing differences modelled here are thus not related to climate, but are inherent to modelled glacier dynamics. They result from differences in the subglacial topography, in particular catchment sizes and hypsometry, of the different outlet glaciers.

For large Alpine glaciers flowing in deep valleys, such as the Rhone, Rhine and Adige Glaciers, several thousand model years are needed to attain a thermodynamical equilibrium between cold-ice advection from the high accumulation areas, upward diffusion of geothermal heat, and heat release from strong basal shear strain. Temperature evolution is, in part, limited by the slow warming of the subglacial bedrock, that has been previously cooled downed by subfreezing air temperature before glacier advance. For larger glaciers, this thermodynamical equilibrium is typically not yet attained around 25 ka. Several Alpine lobes surge and overshoot their balanced extent before thinning and receding towards the mountains as they warm towards thermodynamical equilibrium (supplementary animation). In contrast, peripheral ice caps with basal topography restricted to

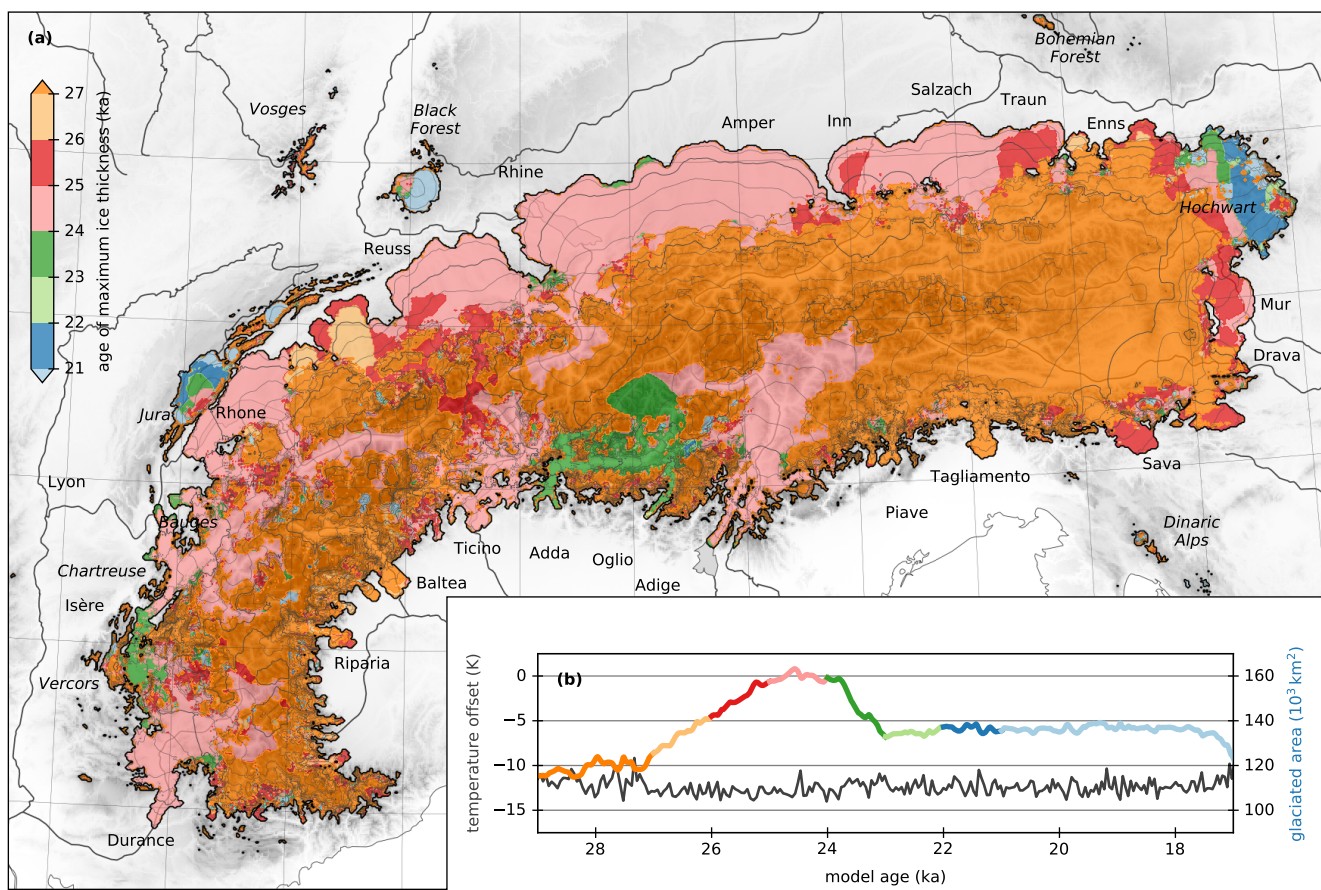

**Figure 5. (a)** Timing of the LGM given by the modelled age of maximum ice thickness throughout the entire simulation (colour mapping) and corresponding, ice surface elevation (200 m contours). **(b)** Temperature offset time-series from the EPICA ice core used as palaeo-climate forcing for the ice flow model (black curve), and modelled glacierized area during the LGM (coloured curve). The LGM is here modelled as a time-transgressive event. In much of the mountain area, maximum thickness is reached before 27 ka.

high elevations experience very low shear strain and virtually no basal sliding. They advance regularly on a frozen bed during the entire cold period, resulting in a late maximum stage (Fig. 5).

Our results indicate that even for a relatively small ice sheet like that found in the Alps, the LGM glacier extent corresponds to a transient stage, while millennial-scale spatial differences in its timing can result not only from climate variability but also from complex glacier dynamics. Heterogeneous climatic anomalies, such as temperature or precipitation anomaly gradients, not included in our model set-up, could perhaps induce further spatial differences in the timing of the LGM Alpine ice sheet, which may counterbalance or enhance those modelled here.

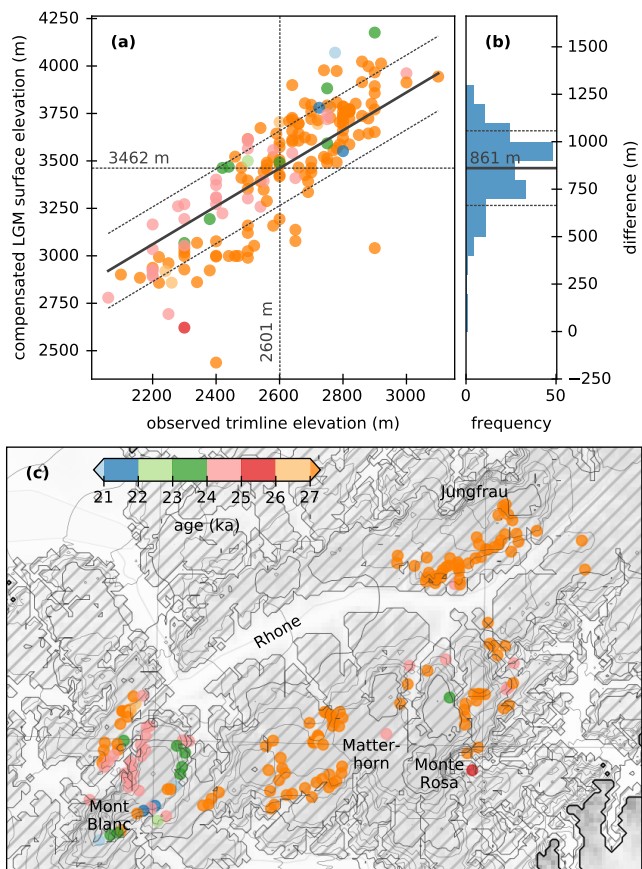

**Figure 6. (a)** Comparison of modelled ice surface elevation at the LGM (time-transgressive, corresponding to maximum ice thickness, Fig. 5), compensated for bedrock deformation, against observed trimline elevations for the upper Rhone Glacier (Kelly et al., 2004, Table 1). Colours show the age of maximum ice thickness (as in Fig. 5). Model variables were bilinearly interpolated to the trimline locations. **(b)** Histogram of differences between modelled LGM ice surface elevation and trimline elevations (50 m bands). The average difference is 861 m. **(c)** Observed upper Rhone Valley trimline locations (Kelly et al., 2004, Table 1), modelled age of maximum ice thickness (colour) and LGM ice surface elevation (200 m contours). Hatches mark the LGM cold-based areas (basal temperature above $1 \times 10^{-3}$ K below freezing at the age of maximum ice thickness).

## 4.4 Ice thickness and trimlines

Following the general assumption that trimlines, which mark the transition between the steep frost-shattered ridges and the more gentle glacially sculpted valley troughs, represent the maximum elevation of the LGM ice surface, maximum ice thickness in the Alps has been reconstructed in several areas (van Husen, 1987; Florineth, 1998; Florineth and Schlüchter, 1998; Kelly et al., 2004; Bini et al., 2009; Coutterand, 2010; Cossart et al., 2012). However, this assumption is challenged by geomorphological and cosmogenic nuclide dating evidence from Scandinavia (e.g., Kleman, 1994; Kleman and Borgström, 1994),

the British Isles (e.g., Fabel et al., 2012; Ballantyne and Stone, 2015), and North America (e.g., Kleman et al., 2010), that pre-glacial landscapes located well above the trimlines have been glaciated and preserved under cold-based ice, sometimes for several glacial cycles (Stroeven et al., 2002). Thus, the trimline could also mark the maximum elevation of the transition from temperate to cold-based ice, or a late-glacial ice surface elevation (Coutterand, 2010, Fig. 1, p. 403). However, no such
evidence has been reported in the Alps. Instead, tree and fire remains record the presence of nunataks in the south-western Alps by 21 ka (Carcaillet and Blarquez, 2017; Carcaillet et al., 2018).

    In the upper Rhone Valley, the maximum ice surface elevation reached during MIS 2 in our simulation, corrected for bedrock deformation, is consistently modelled to have occured several hundred metres above the observed trimline elevations (Fig. 6a and b), with a mean difference of 861 m (and a standard deviation 197 m, Fig. 6b), which is significant in respect to the modelled
maximum ice thickness in the Rhone Valley (and the Alps) of 2586 m. This result depends on uncertain basal sliding and ice rheological parameters to which the model sensitivity was not tested here. However, regional model sensitivity tests show that a modelled surface compatible with the trimline elevations is incompatible with the documented overspilling flow across the Simplon Pass (Becker, 2017; Becker et al., 2017), and LGM palaeo-climate proxy records (Cohen et al., 2018). Unfortunately, validation through the bedrock uplift rate (cf. Kuchar et al., 2012) is not doable in the Alps due its lower values, active tectonics
and uncertainties on geological properties (cf. Mey et al., 2016).

    Our model results depict a polythermal LGM Alpine ice cover. Due to sub-freezing temperatures applied in the climate forcing of the model, the entire Alpine ice sheet is capped by an upper layer of cold ice. In major Alpine valleys, such as the upper Rhone Valley, important ice thickness and strain heating contribute to form a layer of temperate ice near the glacier base, allowing basal melt, sliding and potentially erosion (Fig. 6c, white areas). On the other hand, on the highest mountains, ice
cover is too thin and too static to form temperate ice, resulting in cold ice down to the bed and preventing potential erosion (Fig. 6c, hatched areas).

    In the upper Rhone Valley, observed trimlines are often located near the LGM cold-temperate basal thermal transition or within cold-based areas (Fig. 6c). The remaining discrepancies between the trimlines locations and the modelled basal thermal boundary may relate to temporal migrations of the basal thermal boundary, an absence of sliding in warm-based areas, and
levelling of small-scale topographic features in the 1 km horizontal grid. They call for more detailed comparisons spanning the entire Alpine range and specific sensitivity studies to relevant basal sliding and ice rheological parameters, and to the uncertain subglacial topography. However, the presence of an upper layer of cold ice during the LGM, already found at high altitude in the much warmer climate of today (e.g., Suter et al., 2001; Bohleber et al., 2017), is inevitable (Blatter and Haeberli, 1984; Haeberli and Schlüchter, 1987; Cohen et al., 2018). The bedrock themal model yield significant volumes of frozen ground
periglacially north of the Alps and subglacially under the highest mountains, which is also in agreement with previous studies (Haeberli et al., 1984; Lindgren et al., 2016; Cohen et al., 2018).

    In this context, our results challenge the assumption that Alpine trimlines mark the maximum upper ice surface elevation of the LGM ice cover and call for a more accurate estimation of the thickness of the upper layer of cold ice in the Alps.

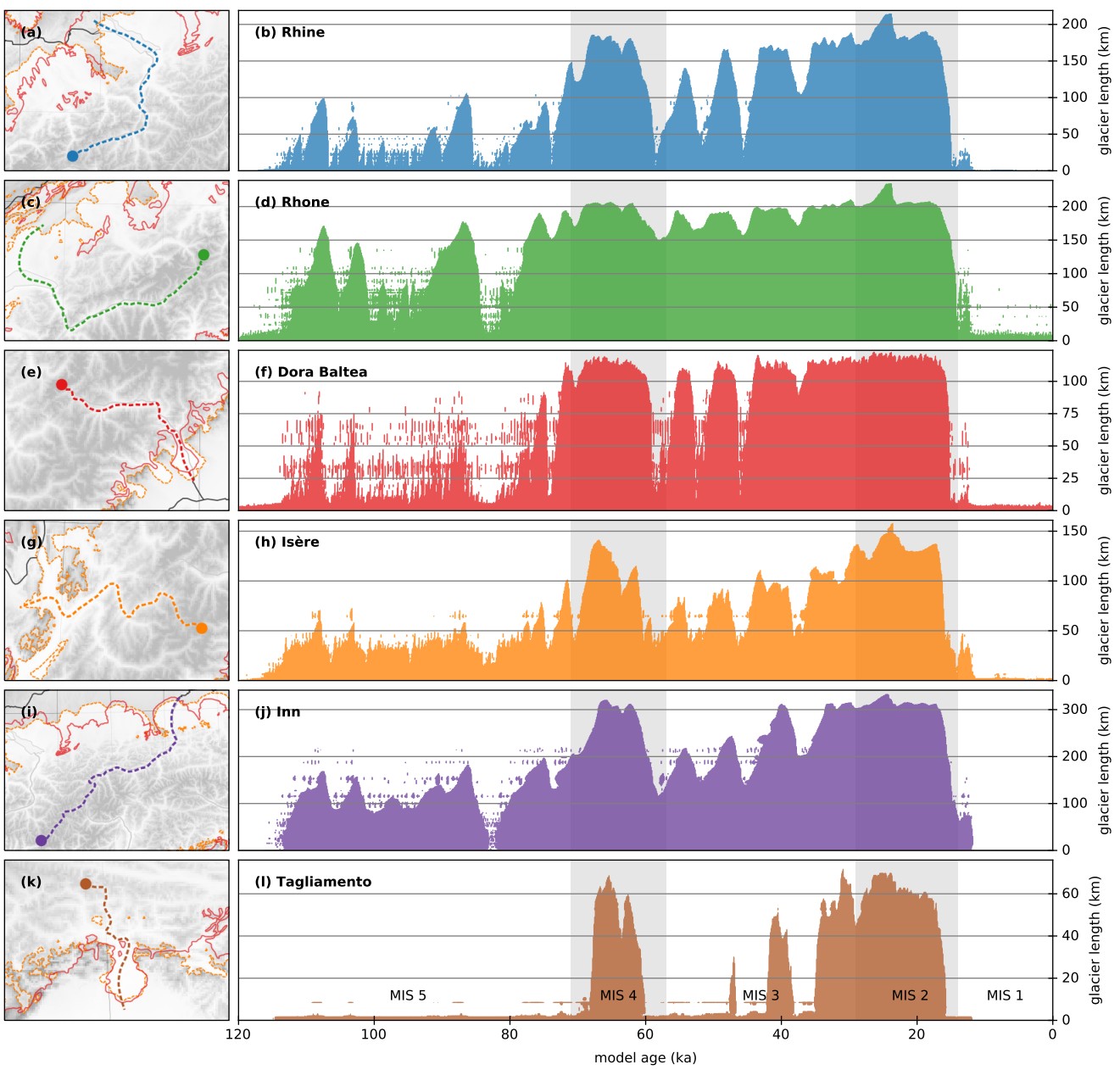

**Figure 7. (a, c, e, g, i, k)** Profile lines roughly following valley centerlines for the Rhine, Rhone, Dora Baltea and Isère, Inn and Tagliamento Glaciers. **(b, d, f, h, j, l)** Evolution of modelled glacier extent in time, bilinearly interpolated along the corresponding profiles, showing numerous cycles of advance and retreat over the last glacial cycle modulated by subglacial topography and catchment geometry. Shaded gray areas indicate the timing for MIS 2 and MIS 4 (Lisiecki and Raymo, 2005). Isolated patches indicate episodic advances from tributary glaciers.

## 4.5 Glacial cycle dynamics

Glacial history of the Alps prior to the LGM remains poorly constrained. Although the four glaciations model (Penck and Brückner, 1909) has long been used, it is now known that glaciers advanced onto the foreland at least 15 times since the beginning of the Quaternary at 2.58 Ma (Schlüchter, 1991; Preusser et al., 2011). Sparse geological data indicate that the last glacial cycle may have comprised two or even three periods of glacier growth and decay (Preusser, 2004; Ivy-Ochs et al., 2008). Luminescence dating from two sites in the central northern foreland indicate an early last glacial advance of Alpine glaciers onto the near foreland around $107 \pm 9$ to $101 \pm 5$ ka (Preusser et al., 2003; Preusser and Schlüchter, 2004). Evidence for a MIS 4 glaciation is equally sparse (Sect. 3.4; Keller and Krayss, 2010; Barrett et al., 2017; Haldimann et al., 2017), and its timing remains uncertain (e.g., Link and Preusser, 2006; Preusser et al., 2007). During MIS 3, major Eastern Alpine valleys hosted wolly mammoths (Spötl et al., 2018) and open vegetation (Barrett et al., 2018), indicating more restricted glaciation.

As previously mentioned (Sect. 3.3), independent of the palaeo-temperature (GRIP, EPICA, or MD01-2444) and palaeo-precipitation (with or without corrections) applied, all simulations presented here result in a high temporal variability in the total modelled ice volume (Fig. 2). Using the optimal EPICA palaeo-temperature record with least variability (Sect. 3.1) and conservatively including palaeo-precipitation reductions (Sect. 3.2), the 1 km resolution simulation also results in strong total ice volume variability throughout the last glacial cycle (Fig. 4b). Two major glaciations occur during MIS 4 and 2 (Fig. 4b). However, several minor glaciations occur during MIS 5 and 3, as well as a minor late-glacial readvance at the onset of MIS 1 (Fig. 4b). These episodes are the result of synchronous advances of several Alpine glaciers well into the major valleys and sometimes even onto the foreland (Fig. 7; supplementary animation).

For instance the Rhine Glacier (Fig. 7a) extends beyond the outermost limestone reliefs six times during the simulation, and sometimes retreats almost completely between two advances (Fig. 7b). The Rhone Glacier, fed by several high-altitude accumulation areas, advances eight times onto modern Lake Geneva (Fig. 7c and d). The Dora Baltea Glacier, characterised by a very steep catchment, and mutliple tributaries, shows even a higher variability and reaches close to its maximum position six times throughout the simulation (Fig. 7e and f). The Isère (Fig. 7g and h) and Inn (Fig. 7i and j) Glaciers, with their complex system of confluences and diffluences, need longer time to build up and reach the foreland only two or three times (Fig. 7g–j). Finally, the Tagliamento Glacier, distant from the major ice-dispersal centres of the inner Alps, is absent for most of the modelled glacial cycle (Fig. 7k and l)

Despite the low temperature variability of the palaeo-climate forcing, and reduced precipitation dampening glacier response, our simulation depicts the Alpine ice complex as highly dynamic, with many more than two or three (cf. Preusser, 2004; Ivy-Ochs et al., 2008) glaciations, and regional glacier dynamics controlled by spatial variations in catchment size and hypsometry. Importantly, Dansgaard-Oeschger events, recorded in Europe (Spötl and Mangini, 2002; Wohlfarth et al., 2008; Luetscher et al., 2015) but absent from the EPICA palaeo-temperature forcing, may have induced an even more dynamic glacier response than that modelled here.

# 5 Conclusions

In this study the numerical ice sheet model PISM (Sect. 2) has been applied to ice dynamics of the last glacial cycle in the Alps. Using three different palaeo-temperature forcing records (GRIP, EPICA, and MD01-2444; Sect. 3.1), scaled to reproduce the Rhine Glacier piedmont lobe in agreement with the mapped LGM ice margin, and two different palaeo-precipitation parametrizations (with and without ca. -68 mm °C$^{-1}$ precipitation reductions; Sect. 3.2), we find that only the EPICA palaeo-temperature record yields model results in agreement with geological findings, in the sense that:

- The EPICA palaeo-temperature forcing yields maximum ice volume at 25.2/24.6 ka (without/with palaeo-precipitation reduction), followed by a standstill of major piedmont lobes in the forelands until 17.3 ka, both compatible with much of the dating results, whereas the GRIP forcing results in early deglaciation of the foreland complete by 21.4 ka, and the MD01-2444 forcing results in a late LGM glaciation peaking at 16.5/15.7 ka (Sect. 3.3).

- The EPICA palaeo-temperature forcing yields cumulative ice extent compatible with geological evidence during MIS 4 and 2, whereas both GRIP and MD01-2444 records result in MIS 4 glaciation well beyond the mapped LGM ice limits (Sect. 3.4).

This interesting result may be coincidental as there is no direct link between European and Antarctic climate. This highlights the need for more quantitative reconstructions of European palaeoclimate.

We then use this optimal palaeo-temperature forcing and, as a conservative approach, palaeo-precipitation reductions, to force a 1-km resolution simulation of the last glacial cycle in the Alps. A more detailed analysis of its output lead us to the following conclusions.

- Ice cover is generally underestimated in the north-western Alps and overestimated in the eastern and south-western Alps, indicating that east-west gradients in temperature or precipitation change, absent from our model forcing, probably controlled the LGM extent of ice cover in the Alps. The observed asymmetric extent of ice north and south of the Alps can be explained by the modelled transient nature of the LGM extent without involving north-south gradients in temperature and precipitation change (Sect. 4.1).

- The LGM ice flow pattern in the Alps was largely controlled by subglacial topography, but transfluences across several mountain passes may have occurred (Sect. 4.2).

- The LGM (maximum) extent was a transient stage in which glaciers were out of balance with the contemporary climate. Its timing potentially varied across the range due to inherent glacier dynamics (Sect. 4.3).

- Ice thickness during the LGM is modelled to be much larger than in reconstructions. In average, modelled surface elevation is 861 m above the Rhone Glacier trimlines, which may instead indicate an englacial thermal boundary (Sect. 4.4).

- Alpine glaciers were very dynamic. They quickly responded to climate fluctuations and some potentially advanced many times over the foreland during the last glacial cycle (Sect. 4.5).

However, these results are limited by uncertainties on glacier physics. The till deformation model used here does not hold for sliding over bedrock surfaces. On the other hand, the constant friction angle used is representative of wet till but weaker basal conditions may have applied over saturated lake sediments where they occured. In the absence of ice deformation measurements, a constant rheology was used for temperature ice containing more than 1 % of liquid water.

More importantly, these results are limited by the simplicity of the surface mass balance parameters and climate forcing used. In particular, additional palaeo-climate variability over the European Alps may have caused more glaciations than modelled here. Shifts in the North Atlantic storm track and polar front may have caused varied patterns of glaciations through different cold phases. Using more specifically targeted sensitivity runs, and a more realistic climate forcing based on regional circulation model output or including the effect of long-term term changes in incoming solar radiation, future modelling studies will certainly be able to quantify uncertainties associated with some of the above limitations. Nevertheless, we hope that these conclusions will also serve as a basis for future studies of glacial geology in the Alps, and call for a more systematic aggregation and homogenization of glacial geological data to form a basis for model validation across the entire Alpine range.

*Code and data availability.* PISM is available as open-source software at http://pism-docs.org. Aggregated model variables including those used to draw figures in this manuscript are available as geotiff (DOI application pending). Selected model output is available as compressed netCDF (DOI application pending). The supplementary animation can be found at https://doi.org/10.5446/35164.

*Author contributions.* J. Seguinot designed the study, ran the simulations, and wrote most of the manuscript. All authors contributed to interpreting the results and improving the text. M. Huss provided modern ice thickness data. The idea for this study stems in part from an excursion organised by F. Preusser in the central Alps in 2012.

*Competing interests.* The authors declare that they have no conflict of interest.

*Acknowledgements.* We are very thankful to Constantine Khroulev, Ed Bueler, and Andy Aschwanden for providing constant help and development with PISM, in particular with recent issues with the computation of ice temperature (Github issue no. 371) and with the computation of bedrock deformation (Github issues no. 370 and 377). We are equally thankful to three anonymous reviewers, Giovanni Monagato, and the editor Andreas Vieli for their meticulous readings and constructive inputs during peer-review. We thank Marc Luetscher for an insightful discussion of preliminary results during the EGU General Assembly 2017. The experimental design and article layout used here are based on previous work which received much guidance from Irina Rogozhina and Arjen P. Stroeven. The current work was supported by the Swiss National Science Foundation grants no. 200020-169558 and 200021-153179/1 to M. Funk. Computer resources were provided by the Swiss National Supercomputing Centre (CSCS) allocations no. s573 and sm13 to J. Seguinot.

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
