# Peer review of "Modelling last glacial cycle ice dynamics in the Alps"

_The Cryosphere, 2018_

## Short Comment (SC1) · 23 Mar 2018

The manuscript is very interesting as approach on the Late Pleistocene Alpine glaciation, for which good data are available for the LGM onwards, but few is known about pre-MIS2. The present article does not solve the problem of what how the glaciers behave before the LGM, but casts new light on this topic suggesting interesting details and field to investigate. I would like to post some general comments on the manuscript especially regarding the Italian side of the Alps.

1- Reading the manuscript and watching the supplemental file, it is remarkable how few systems in the model result to have the same extension compared to the geomorphological/geological evidence. For the Italian side roughly the Riparia, Baltea, Ossola/Ticino and Tagliamento looks right. While other systems are much over-(western Alps) or under-estimated (from Adda to Piave). The overestimation of the western Alps can be related to the scarcity of updated chronological data; for example the glacier reconstruction in Gesso Valley of the Maritime Alps (Federici et al. 2017) shows a much larger extent than Ehlers and Gibbard (2004) compilation. Anyhow the sentence of line 30 page 14 is reliable and suggests that more data are needed about. Also the Eastern Alpine glaciers resulted very overestimated, for these and for the not-matching glaciers in the Italian side I think that paleoclimate forcing and especially precipitation models are one of the keys factors. The WorldClim model that is considered and showed in Figure 1 seems to be inconsistent respect to other models focused on the Alps. I suggest to take into consideration Isotta et al. (Int. J. Climatol. 2013) where distribution of the precipitations shows areas of high precipitation rates (both as annual mean and daily peak). This distribution would point to high precipitation rates also in the Piave, Adda and Oglio catchments, while the Tagliamento and Ticino systems fit well in the model as well. The knot of the Valais-Ticino-upper Rhine has also modern high precipitations, and this is one of the key areas for large ice accumulation and for the southerly component during the LGM according to Luetscher et al. (2015). Actually, considering the modern precipitation rates and comparing to your model, the underestimation for the Adige system sounds reasonable but not in agreement with the chronology and geomorphology found in the Garda. So other causes, and not only precipitation, have to be considered.

2- Concerning the ice-transfluence. I agree that it has to be much more considered. I wonder if the Adige had a great contribution from the Austrian Tauern (Toblach area) this would have increased the ice in Adige and deplete the Drava, or in Winschgau valley where the catchment upstream Mustar saddle is more related to Adige than the Inn. The same could have happened for the Piave catchment, which is very underestimated in the model. Again, ice flowing to the south did not flow to the east. But I think that this may be not enough for justifying the overestimation of the Eastern Alps. Concerning the Straninger saddle I am a bit skeptic because it is not the lowermost saddle
(Plockenpass and Nassfeldpass are lower in elevation). The central Adda glacier (and not the Ticino as at page 14 line 26) is much underestimated and here the contribution from transfluence in St. Moritz area could have been remarkable. I attach a figure separately with transfluences (yellow stars and red arrows).

3- The basal sliding of glaciers and its overall velocity is another interesting factor to discuss. For Adige for example we know that at about 28 cal BP the Adige glacier was damming a tributary valley north of Trento around the same age (Avanzini et al 2009), while the same glacier arrived at Garda at 24.6 cal BP (Monegato et al., 2017). This means that the glacier front advanced of about 100 km in around 3.5 ka so 280 m/y. The same could have been for the Adda and Oglio glaciers, even if a such robust chronology is lacking. Is it possible that overestimated velocities produced large eastern glaciers?

4- About the self-sustained ice domes, their importance is for me unclear and not well explained in the text. Why they are only two? Why the Adamello or the Tauern massifs, as an example, were not considered as self-sustained ice domes?

5- Global circulation models (e.g., Löfverström et al., 2014, Beghin et al., 2015) suggest that the Polar front was at different latitudes during each cold phases of the Late Pleistocene. This could have effect of different impact on the Alps. For example during MIS4 if the westerlies were dominant this could have driven more effective precipitations in the western and northern Alps in respect to the southern and the eastern Alps. If this is true, and can be applied to the major cold phases. How was the behavior of glaciers during MIS 5 and 3?

Please also note the supplement to this comment: https://www.the-cryosphere-discuss.net/tc-2018-8/tc-2018-8-SC1-supplement.zip

---

## Referee Comment (RC1) · Anonymous Referee #1 · 16 Apr 2018

This paper is a landmark advance in modelling European Alpine ice cover, applying a high-resolution (1 km) ice model to the entire Alps through the last glacial cycle, for the first time to my knowledge. Results are compared with diverse geological data, and several important findings are presented, including time-transgressive ice marginal extents at LGM. The climate forcing is simple, applying uniform perturbations to modern observed datasets, which leads to some uncertainty in the results, but does not detract from them too much given the advances made in the ice modelling alone.

The introduction gives an elegant summary of Alpine glacial science since the 1700's, including many historical references. The paper is well organized, with well-chosen sensitivities described first that calibrate the climate forcing, followed by detailed analysis of one best-fit high-resolution (1 km) simulation through the last 120 kyrs. Detailed

comparisons to a variety of geological data are made, constituting a thorough assessment of model performance. An impressive animation of the whole cycle is included as supplementary material.

Specific comments:

pg. 4, lines 9-10: Can the physical basis of englacial water fraction and sensitivity of results be summarized briefly? This is not a usual component in ice-sheet models. Is the cap value ("capped at 0.01") well constrained, and does it have a significant effect on results?

pg. 4, lines 19-20: The sub-glacial hydrologic component should be described more (even if exactly as in Bueler and van Pelt, 2015). Is basal water transported horizontally down the hydropotential gradient? This is usually a highly uncertain component of ice-sheet models, but can have a large effect on results through its influence on basal sliding, and basal frozen vs. thawed areas, which is relevant to section 4.4 regarding trimlines.

Somewhat related: Little information is given on the choices of basal sliding parameter values in Table 1. This could be discussed briefly. Presumably no inversion or optimization was performed for these values beforehand, and they do vary spatially. Are they appropriate for Alpine bedrock overall?

pg. 7, line 7: Are there any data to support this atmospheric lapse rate value (6 K km-1), and do other values have the potential to significantly affect ice temperatures? In particular, could they change the basal areas of frozen/unfrozen ice and so the comparisons with trimlines in section 4.4?

Climate forcing:

The method of spatially uniform shifts to modern climate forcing is common in paleo-modeling of large ice sheets, and in my opinion is acceptable as a starting point in this work, with coupling to regional climate models (RCMs) left to follow-on work. There

are good discussions on possible shortcomings of this method, for instance as a cause of anomalous east-west marginal ice extents at LGM (pg. 11, line 6-8). However, I suggest changing the sentence on pg. 9, line 5, which mentions some RCMs applied to LGM Europe, but also says "...over the Alps during the last glacial cycle, of which little is known apart from the LGM". There are several other RCM modeling studies over Europe during the last 120 kyrs, e.g., for MIS Stage 3, Kjellstrom et al., Boreas, 2010; Barron and Pollard, Quat. Res., 2002; Alfano et al., Quat. Res., 2002; and for 6 ka, Strandberg et al., Clim. Past, 2014. Perhaps there is little useful material there for the Alps, but such papers exist.

The past climate variations are prescribed following 3 quite distal core records, and the most distal (EPICA) is chosen as yielding the best fit to Alpine glacial evidence. The basis for preferring EPICA seems reasonable (matching some higher-frequency amplitudes of ice variability, section 3.3). However, this agreement is in a sense co-incidental, in that there is no direct meteorological link between Antarctic and Alpine regional climate variations. Are there any proximal proxy records of Alpine climate at all, perhaps lacustrian varves, that could be used to assess the EPICA-based shifts in air temperatures and precipitation, even over limited periods of the last 120 kyrs?

A PDD scheme is used based only on seasonal air temperatures. van de Berg et al. (Nature Geosc., 2011) showed that for long-term variations including the Eemian, orbital changes in insolation are important and should be considered explicitly. This could be particularly relevant here, because the EPICA core does not reflect changes in insolation over the Alps. In further work, an insolation-change term (summer, local) could be combined with EPICA in the climate temperature paleo-forcing.

Trimlines:

The point is well taken that trimlines do not necessarily indicate past ice surface elevations, but the upper limit of temperate ice with cold ice above (pg. 18, line 5 to pg. 19, line 4). It is an important point, because model LGM ice surfaces are far above most

trimline elevations (as noted and in Fig. 6a,b). The pertinent results are shown in Fig. 6c, and I agree there is good support for the cold vs. temperate (basal) ice hypothesis.

It might help general readers to spell out the interpretation even more in the text. That is, as I understand it, the observed trimlines should coincide with the boundaries between model areas of frozen vs. temperate beds, so the dots in Fig. 6c should all lie on the borders between the hatched and white areas. The sentence on pg. 19, lines 20-21, is confusing in this regard. Incidentally, it would also help to add the word "basal" to the last sentence of the Fig. 6c caption: "...experienced temperate basal ice for ...".

One reason for the remaining discrepancies in Fig. 6c could be temporal variations in the model boundaries, that are aggregated in time by the "< 1 kyr" criterion for the hatching and the grouping of all trimline data. To go into this in more detail, in principle Fig. 6c could be expanded to show the model basal frozen-temperate boundaries at particular times (21.5, 22.5, etc, ka), with only the dots for each time period superimposed. But that may not be worth it unless there are large temporal variations in the model boundaries.

A slight concern is that the majority of the trimline data seems to be orange dots i.e., older that 27 ka in the timescale of Fig. 6c. The period for the model hatching extends back only to 29 ka. Hopefully, most of the orange-dotted data are within that period and are not older than 29 ka(?).

The text could briefly mention (and hopefully rule out) the issue of very fine-scale topographic features on which the trimlines are located, not resolved by the 1-km model topography. If data sites are on small-scale highs or lows significantly different from their ∼km-scale surroundings, that could contribute to the discrepancies in Fig. 6c.

Technical comments:

pg. 3, line 2: Perhaps change "lead" to "led", "to which" to "to what".

Fig. 1 caption, line 4: Perhaps change "estimated" to estimate", or "estimated of" to "estimated".

[Figure]

---

## Referee Comment (RC2) · Anonymous Referee #2 · 20 Apr 2018

Review of manuscript "Modelling last glacial cycle ice dynamics in the Alps" by Julien Seguinot, Guillaume Jouvet, Matthias Huss, Martin Funk, Susan Ivy-Ochs and Frank Preusser, Cryopshere Discuss., https://doi.org/10.5194/tc-2018-8

General comments

This manuscript describes a model study with the Parallel Ice Sheet Model applied to the last glacier cycle in the Alps. The climate forcing is derived from present day climate of WorldClim and the ERA-Interim reanalysis and time-dependent temperature offsets derived from the Greenland Ice core (GRIP), from Antarctic ice core (EPICA) and Marine sediment core from the Iberian margin (MD01-2444). The study is split in two, in the first part analysis of six model simulations (with and without precipitation scaling) made on a 2 km resolution grid is presented and concluded that out of the three climate forcing records used, the EPICA record gives the most realistic ice volume history during MIS 4 and 2. The second part analyses simulation made with the EPICA forcing record on a 1km grid.  The authors draw conclusions about the ice cover, ice flow pattern, ice thickness, LGM ice extent, which in their model is a transient stage with varying timing across their model domain due to glacier dynamics.  The manuscript is well organized and clearly written, the missing thing in this study is a discussion of and preferably a sensitivity study of the ice dynamic-model assumptions made.  Is the sliding of the ice age ice sheet realistically modelled with pseudo-plastic assumption using the Shallow Shelf approximation?  How sensitive are the results to the selected model parameters?  It is briefly mentioned once on page 19, line 8, but a thorough analysis of the model sensitivity would strengthen the paper.

Specific comments:

I find missing something that indicates that the times are before present, as in line 5, line 8 and line 16 on page 1 – and elsewhere in the paper  it is written (120-0 ka)  should you add before present or BP to indicate the time interval?

Could the reason for too large ice volumes when using the GRIP record, as mentioned in lines 13-15 on page 9 be due to Arctic Amplification? Could that have an effect then as it has now? This is also mentioned in line 11 on page 10

Minor comments:
Page 1 line 24 suggest: "have extended well outside their current margins"
Page 2 line 22, could you add a reference and a timing for LGM?
Page 3, line 1, suggest to replace "thus" with "still"
Page 3, line 3  add s to responses
Page 3, line 18, something missing in the sentence, suggest "formulation" after "creep"
Page 4, line 4 suggest to replace "field" with "sheet"
Page 5 figure 1 c) can you add a scale and maybe indication with a box in b) where this extract is from? "from the estimate" (not plural in line 4 of caption),
Line 6 (PDD) is not acronym for surface mass balance, some more explanation is needed here,  indicate also, like in the other figures that h) is January and i) is July figures

Page 6 line 6 suggest to replace "of" with "with"

Page 6, line 14, suggest "The climate forcing driving the ice sheet simulations consist spatially of a present-day monthly mean climatology.. "

Page 6 line 15 suggest to delete "s" on corrections

Page 6 line 19 add "mean" after monthly

Page 6 line 22 note, if true (clarify in Figure caption, see comment above) then the reference should be to Fig. 1 i) for summer precipitation

Page 6 line 27 replace "is" with "are"

Page 6, footnote, add a reference for the correct formula for the rigidity and clarify (add something like, the consequence of this error is that the simulations effectively use….)

Delete "in" before "a small" and add a quantification of the small change in the length scale – how small, is it a few percentage?

Page 7, line 2 "shipped with WorldClim" is not clear, please edit, also suggest not to use b for topography, $s_{surf}$ and $s_{bed}$, or h would be better

Page 7 line 19 add "the" before "oxygen"

Page 7, line 24 "and within a rectangular region .." is not clear, edit this text

Page 8   add "acceleration of" before gravity, add a reference for the ideal gas constant?

Page 9 line 14-15, see comment above, could this be an example of Arctic amplification?

Page 9 line 19, add "a" before "very"  and suggest to turn the sentence around, the EPICA simulations are in a good agreement with the data

Page 9, line 20, followed by first a retreat and then a standstill?  replace "blue" with "red"

Page 9, line 21, suggest : The simulations forced by the GRIP palaeo-temperature forcing yield … (blue curves)"

Page 10, line 2, suggest to add "followed by a rapid retreat"

Page 10, line 3, why state "two or three" in intro the suggestion is either 4 major or 15, is there a reference for 2 or 3?

Page 10, line 10, suggest to replace "lower" with "smaller"

Page 10, line 11, is this due to Arctic Amplification?

Page 10, line 12, suggest to replace "least" with "the smallest"

Page 11, figure 3 caption, what does "cumulative" mean here? Do you mean maximum in each location? clarify what the black line indicates. No solid red line is visible in figures. What is meant with "reasonable", maybe replace with "realistic"?  suggest to replace "cover" with "extent"

Page 11, line 2, suggest to replace "cumulative" with "maximum extent in each area" or something similar

Page 11, line 1-3, how sensitive is the model to different parameters in the applied sliding formulation?

Page 11, line 4-5 "outside this benchmark" clarify if you mean spatially or temporally

Page 11, lines 6-11 is this text better fitted in a discussion section?

Page 12, line 2, suggest to replace "higher" with "larger"

Page 12, line 4, do you mean to refer to Fig. 3?

Page 12, line 29-32 does this text fit better in method section?

Page 13, figure 4 caption, the 200 m surface contours are not clearly visible in the figure, can they be made sharper, or just skipped? Line 3, something like "are shown" is missing. Suggest to replace

"background" with "bedrock". This is the first time "Natural Earth Data" is mentioned, should that be in the section on the data?  Suggest to replace "Gray fields" with "shaded gray area" and "boundaries" with "timing"

Page 15, line 8, replace "was" with "is"

Page 15, line 25  "occurred"

Page 16 figure 5, in figure the color bar is written to indicated maximum surface elevation, but in figure caption the maximum ice thickness, which is correct?  The surface elevation contours are not clearly visible.  The dark orange color in the figure, that covers the central part of the ice sheet is not (?) in the bar on the left (or is it before 27 ka BP?)

Page 17, line 30 suggest to replace "have" with "could"

Page 18, figure, the 200 m contours are not clearly visible

Page 19, line 7, could you add what the modelled regional ice thickness is, for comparison?

Page 20, add info about the gray areas indicating MIS 4 and 2  "Isolated patches indicate periodic surges from tributary glaciers" needs more explanation and it is not clear what is meant.  Does the model simulate periodic surges?

Page 21 line 21, here is a reference for 2 or 3 glaciations (see comment above) but in the intro is only mentioned 4 or 15, suggest to change text to harmonize.

Page 21 line 27, suggest to edit "the study consists of" or start for example like "In this study the model has been applied…"

Page 21 line 28, how important is it that the model has been validated for the Cordilleran ice sheet?  Will that support the choices of the sliding model applied in the Alps?  I think that it would be valuable for this study to do a sensitivity runs for at least some of the model parameter choices.

Page 22, line 6, add "records" after forcing

Page 22, line 19, why do you add "potentially" here? Isn't this a firm conclusion from you study?

Page 22, line 20, suggest to replace "higher" with "larger"

Page 22, line 22, same as above, why "potentially" here?

Page 22, line 24, suggest to replace "nevertheless" with "however" or edit the sentence

Page 22, line 26-28, the paper would be able to give stronger conclusions with sensitivity study, suggest to edit the sentence by replace "statements" with "limitations" or "drawbacks"  and "last glacier cycle ice dynamics in the Alps" is not easy to read

Page 23, line 2, replace "mode" with "more"

Figures are generally clear and well set up.  The surface contours in Figs 4,5 and 6 is not clearly visible in my printout and could maybe become clearer?

---

## Referee Comment (RC3) · Anonymous Referee #3 · 23 Apr 2018

This is an excellent manuscript and I highly recommend publication after a few minor changes. The authors present a well thought out modelling experiment which they combine (albeit in a qualitative manner) with extensive palaeo-glaciological data and cumulative work. I can see this work being extended into more extensive and rigorous work (RCM forcing, ice dynamic sensitivity, quantitative fitting to geomorphological record etc), but this is an important leap forward.

Comments:

Abstract

P1, L2: "pioneer" should be "pioneering" P1, L16: I think the finding that you get asynchronous glaciation extents with a uniform climate offset is due to glacier hypsometry

and setting should be stated here. i.e. that the timing of maximum glaciation and recession isn't purely a function of climate. This finding needs to be highlighted better in the abstract.

Introduction

P2, L28: Ballantyne and Stone (2015) should be added to this list. P2, L34: It should be stated that it could be a consequence of both glaioclimatic interactions and uncertainties in dating methods. P3, L2: These points serve the literature well to highlight gaps for future research. However, I would argue that you do not get very far here on 1 and 5 and do not completely solve the other 3 points. Your text reflects these shortcomings very well, for which you should be applauded. Though I think at this stage of the manuscript, your statement of intent, you should state that you do not claim to solve these questions, but rather push forward on all of them using your new approach of ice sheet modelling.

Section 2.6 P7: The spatial distribution of your modern climate variables (precip, temp) will be massively influenced by elevation. Though there is a lapse rate, does this pattern of high precip and low temp over mountains remain throughout the simulation despite ice surface topography, and if so, how does this influence your results?

Section 3: P7, L14: You keep mentioning the number of processors. I find this information slightly irrelevant, and it will soon become outdated as processing speed and models increase (GPUs for example). The only way it will serve the community is if there is a full description of the computer set up. For example, it could be that the simulations took 4 days on 144 processors, but the processors were slow. Suggest removing these references.

P10, L14: This is a more general point. You eventually choose the EPICA record, and for justifiable reasons based on comparison to reconstructed ice extent and timing. However, this is likely coincidence. EPICA is likely a complex record containing global and local antarctic influences upon climate. The "real" climate over the alps during

glaciation is like decoupled from that of Antarctica to an extent. Therefore, the match you find is not an inference about climate, as different combinations of offsets may have made the same result (smoothed GRIP to remove some of the D-O scale noise?). You should make this explicit somewhere in the manuscript.

P11, L2: First sentence needs reconsidering as it is slightly broken in its current form. Perhaps "Figure 3a shows the cumulative extent of glaciated area during MIS2".

P11, L10: Is it possible some ice is missing from the geological reconstruction in some instances? I guess some outlets are well constrained, whilst others areas could be "filled in" by this modelling experiment.

Section 4

P14, L15 - 31: I find this description of sites and timing of glacier extent compared to dates difficult to follow. I suggest a new figure to convey this important comparison: Have the reconstructed and modelled ice extents at key times for each of the mentioned glaciers on several smaller maps, including geochronological constraints.

P17, L17: This finding is important and should be highlighted better in abstract and conclusion.

P17, L22: Is the model recreating possible surging? Or would this be an over-interpretation given the uncertainty in climate and physics. Seems to fit with the enthalpy model of Benn and others for surging. As reads, it suggests that these areas were possible palaeo-surges, please clarify.

P19, L9-12: On trimlines: I think you justify well why you haven't yet modelled the sensitivity to your trimline result - previous work backs this up. But, I think you should consider the following: Adding a plus/minus to your results to reflect the uncertainty. The importance of resolution - many trimlines will be below the resolution of your model, so perhaps aren't resolved enough for model-data comparison. They may have acted to deflect ice flow around mountain peaks for example. This resolution caveat should be

mentioned. I would be surprised if all trimlines are subglacial transitions as this paper suggests - perhaps you need to directly challenge the geochronological community to find better trimline constraints (sub/supra) as a statement in this paper. Your mean value of 861 m is unrepresentative of your sample. Your sample is highly skewed, so a modal value (1050 m ish from Fig 6) is more appropriate. A similar finding with a similar approach has already been found for the British Isles, with the added constraint of GIA observations. I suggest referencing Kuchar et al. 2011 for this reason.

A philosophical but important point is that your discussion throughout is written from the standpoint of the geochronological/geomorphological data and reconstructions as being "truth". It should consider somewhere that perhaps data is missing as it is hard-won in the places it exists and interpretations may be slightly wrong. The model is also not the "truth" and there is probably a blurred line inbetween upon which we can proceed.

P21, L11-19: Be really clear here that these are modelled, and perhaps not geologically recorded, advances of the ice sheet. If there is no data, it might be correct, might be just a modelled result.

Additional references: Ballantyne, C.K. and Stone, J.O., 2015. Trimlines, blockfields and the vertical extent of the last ice sheet in southern Ireland. Boreas, 44(2), pp.277-287. Kuchar, J., Milne, G., Hubbard, A., Patton, H., Bradley, S., Shennan, I. and Edwards, R., 2012. Evaluation of a numerical model of the British–Irish ice sheet using relative sea‐level data: implications for the interpretation of trimline observations. Journal of Quaternary Science, 27(6), pp.597-605.

---

## Author Comment (AC1) · 23 Jul 2018

Dear Giovanni Monegato,

Thank you very much for your public comment on our manuscript.

The manuscript is very interesting as approach on the Late Pleistocene Alpine glaciation, for which good data are available for the LGM onwards, but few is known about pre-MIS2. The present article does not solve the problem of what how the glaciers behave before the LGM, but casts new light on this topic suggesting interesting details and field to investigate. I would like to post some general comments on the manuscript especially regarding the Italian side of the Alps.

Thank you very much for your supportive comments!

**1 –** Reading the manuscript and watching the supplemental file, it is remarkable how few systems in the model result to have the same extension compared to the geomorphological/geological evidence. For the Italian side roughly the Riparia, Baltea, Ossola/Ticino and Tagliamento looks right. While other systems are much over-(western Alps) or under-estimated (from Adda to Piave). The overestimation of the western Alps can be related to the scarcity of updated chronological data; for example the glacier reconstruction in Gesso Valley of the Maritime Alps (Federici et al. 2017) shows a much larger extent than Ehlers and Gibbard (2004) compilation. Anyhow the sentence of line 30 page 14 is reliable and suggests that more data are needed about. Also the Eastern Alpine glaciers resulted very overestimated, for these and for the not-matching glaciers in the Italian side I think that paleoclimate forcing and especially precipitation models are one of the keys factors. The WorldClim model that is considered and showed in Figure 1 seems to be inconsistent respect to other models focused on the Alps. I suggest to take into consideration Isotta et al. (Int. J. Climatol. 2013) where distribution of the precipitations shows areas of high precipitation rates (both as annual mean and daily peak). This distribution would point to high precipitation rates also in the Piave, Adda and Oglio catchments, while the Tagliamento and Ticino systems fit well in the model as well. The knot of the Valais-Ticino-upper Rhine has also modern high precipitations, and this is one of the key areas for large ice accumulation and for the southerly component during the LGM according to Luetscher et al. (2015). Actually, considering the modern precipitation rates and comparing to your model, the underestimation for the Adige system sounds reasonable but not in agreement with the chronology and geomorphology found in the Garda. So other causes, and not only precipitation, have to be considered.

Thank you for this detailed analysis. We want to point out that WorldClim is not a model, but is also based on observations. However we agree, that the regional dataset by Isotta et al. (2013), tailored to the dense station network and steep topography of

the Alps, is certainly a more robust precitation forcing that needs to be considered in future studies. A reference was added:

> *Finally, modern precipitation data from WorldClim also bear uncertainties and exhibit local disagreement with other regional data (Isotta et al., 2013).*

Besides, the following sentence was added to the discussion of glacier extent regarding the south-western Alps,

> *However, geochronological data from the south-western Alps are sparse, and the LGM extent compilation (Ehlers and Gibbard, 2004) is inconsistent with more recent regional reconstructions (Federici et al., 2017).*

**2 –** Concerning the ice-transfluence. I agree that it has to be much more considered. I wonder if the Adige had a great contribution from the Austrian Tauern (Toblach area) this would have increased the ice in Adige and deplete the Drava, or in Winschgau valley where the catchment upstream Mustar saddle is more related to Adige than the Inn. The same could have happened for the Piave catchment, which is very underestimated in the model. Again, ice flowing to the south did not flow to the east. But I think that this may be not enough for justifying the overestimation of the Eastern Alps. Concerning the Straninger saddle I am a bit skeptic because it is not the lowermost saddle (Plockenpass and Nassfeldpass are lower in elevation). The central Adda glacier (and not the Ticino as at page 14 line 26) is much underestimated and here the contribution from transfluence in St. Moritz area could have been remarkable. I attach a figure separately with transfluences (yellow stars and red arrows).

The amount of ice flowing through transfluences is certainly dependent on model resolution and basal sliding parameters. However, we agree that this will have little effect

on the ice volume overestimation in the eastern Alps in comparison to potential short-comings in the climate forcing.

The tranfluence at the Straniger Saddle may be a byproduct of bedrock topography aggregation to the 1 km grid. As a follow-up study, we have begun to use higher model resolution to study the location of transfluences in more detail (Seguinot et al., 2018). In these results, the transfluence at the Straniger Saddle looses importance relative to other transfluences in the region including, in fact, Plöckenpass and Nassfeldpass. Because the choice of transfluences plotted on Fig. 4 is somewhat arbitrary, we have decided to update it.

**3 –** The basal sliding of glaciers and its overall velocity is another interesting factor to discuss. For Adige for example we know that at about 28 cal BP the Adige glacier was damming a tributary valley north of Trento around the same age (Avanzini et al 2009), while the same glacier arrived at Garda at 24.6 cal BP (Monegato et al., 2017). This means that the glacier front advanced of about 100 km in around 3.5 ka so 280 m/y. The same could have been for the Adda and Oglio glaciers, even if a such ro-bust chronology is lacking. Is it possible that overestimated velocities produced large eastern glaciers?

Thank you. In fact basal sliding is one of the major uncertainties in our results which is now better acknowledged for in our manuscript following other reviewer's comments. However, basal sliding velocity are somewhat decoupled from glacier front advance ve-locities and glacier front maximum extent. Although this is not shown in our manuscript, increased basal velocities tend to result in thinner ice tongues. Due to the mass-balance-elevation feedback, this causes more surface melt and, in turn, lesser extent. However this effect is small in comparison to changes due to climate forcing.

**4 –** About the self-sustained ice domes, their importance is for me unclear and not well explained in the text. Why they are only two? Why the Adamello or the Tauern massifs,

*as an example, were not considered as self-sustained ice domes?*

By self-sustained ice domes, we mean ice domes that are not sitting on top of basal topographic highs, but instead shift away from the modern topographic divides and "sustain" themselves by reaching higher elevation than the local mountains. We indicate these as a sign that ice begins to behave as in an ice sheet, and flow in directions independent from the local basal topography.

However, we agree that this is somewhat arbitrary, and again, as we found out, dependent on model resolution (Seguinot et al., 2018). Therefore, we decided to remove the "ice domes" from Fig. 4a and the main text, and save this discussion for future studies.

**5 –** Global circulation models (e.g., Löfverström et al., 2014, Beghin et al., 2015) suggest that the Polar front was at different latitudes during each cold phases of the Late Pleistocene. This could have effect of different impact on the Alps. For example during MIS4 if the westerlies were dominant this could have driven more effective precipitations in the western and northern Alps in respect to the southern and the eastern Alps. If this is true, and can be applied to the major cold phases. How was the behavior of glaciers during MIS 5 and 3?

We agree. In fact, our simulations are limited by the constant temperature and precipitation patterns applied throughout the glacial cycle. Although we can't answer this question without new climate and glacier simulations, we have raised this important point in the very last paragraph:

> *Shifts in the North Atlantic storm track and polar front may have caused varied patterns of glaciations through different cold phases.*

With this, we would like to thank you again very much for the time and effort you have put into our manuscript.

**References**

Ehlers, J. and Gibbard, P. L., eds.: vol. 2 of *Dev. Quaternary Sci.*, Elsevier, Amsterdam, 2004.

Federici, P. R., Ribolini, A., and Spagnolo, M.: Glacial history of the Maritime Alps from the Last Glacial Maximum to the Little Ice Age, Geol. Soc. Spec. Publ., 433, 137–159, doi: 10.1144/SP433.9, 2017.

Isotta, F. A., Frei, C., Weilguni, V., Tadić, M. P., Lassègues, P., Rudolf, B., Pavan, V., Cacciamani, C., Antolini, G., Ratto, S. M., Munari, M., Micheletti, S., Bonati, V., Lussana, C., Ronchi, C., Panettieri, E., Marigo, G., and Vertačnik, G.: The climate of daily precipitation in the Alps: development and analysis of a high-resolution grid dataset from pan-Alpine rain-gauge data, Int. J. Climatol., 34, 1657–1675, doi:10.1002/joc.3794, 2013.

Seguinot, J., Ivy-Ochs, S., and Imhof, M.: A database of Last Glacial Maximum transfluences and crosswise divides based on sub-kilometer Alpine ice flow modelling., in: EGU General Assembly Conference Abstracts, vol. 20, p. 16268, 2018.

---

## Author Comment (AC2) · 23 Jul 2018

Dear Anonymous Referee #1,

Thank you very much for your detailed review of our manuscript.

This paper is a landmark advance in modelling European Alpine ice cover, applying a high-resolution (1 km) ice model to the entire Alps through the last glacial cycle, for the first time to my knowledge. Results are compared with diverse geological data, and several important findings are presented, including time-transgressive ice marginal extents at LGM. The climate forcing is simple, applying uniform perturbations to modern observed datasets, which leads to some uncertainty in the results, but does not detract

from them too much given the advances made in the ice modelling alone.

The introduction gives an elegant summary of Alpine glacial science since the 1700's, including many historical references. The paper is well organized, with well-chosen sensitivities described first that calibrate the climate forcing, followed by detailed analysis of one best-fit high-resolution (1 km) simulation through the last 120 kyrs. Detailed comparisons to a variety of geological data are made, constituting a thorough assessment of model performance. An impressive animation of the whole cycle is included as supplementary material.

Thank you very much for these elogious and supportive comments!

**Specific comments**

**p. 4, l. 9–10:** Can the physical basis of englacial water fraction and sensitivity of results be summarized briefly? This is not a usual component in ice-sheet models. Is the cap value ("capped at 0.01") well constrained, and does it have a significant effect on results?

Thank you for bringing this up. Laboratory experiments have demonstrated that the rheology of temperate, polycrystalline ice depends on its content in liquid water (Cuffey and Paterson, 2010, p. 65–66). Unfortunately, the only measurements available to date (Duval, 1977), used to quantify the effect of liquid water on ice softness, the creep parameter $A$ in Glen's flow law (Lliboutry and Duval, 1985), only extend to fractions of liquid water content between 0 and 0.8%. They show a three-fold increase of ice softness over this range (Duval, 1977, Fig. 1).

Ice sheet models have previously ignored this effect, but it has now become a typical component of polythermal models such as SICOPOLIS (Greve, 1997), COMICE (Rückamp et al., 2010), PISM (Aschwanden et al., 2012), ISSM (Seroussi et al., 2013), and TIM-FD³ (Kleiner and Humbert, 2014).

However water fractions between 1 and 5 % have repeatedly been observed in temperate glaciers (Murray et al., 2000, 2007; Bradford and Harper, 2005; Bradford et al., 2009), and also occur in model results (e.g., Blatter and Greve, 2015), but it is not known whether ice viscosity continues to decrease substancially for values above 0.8 %. Previous modelling studies have commonly assumed constant ice viscosity above 1 %. This arbitrary threshold is not constrained at all, and the urgent need for new ice deformation experiments has already been pointed out (Kleiner et al., 2015).

The sensitivity of our results to the 1 % threshold was not tested. However, in our model results, liquid water fractions above 1 % typically only occur within the basal temperate layer of the fastest-moving glaciers where ice movement is largely dominated by basal sliding. We suspect that increased ice deformation in these regions is negligible in comparaison to uncertainties related to basal sliding and, more importantly, to climate forcing.

Thus, we prefer to avoid including the above discussion in the manuscript, but have reworked the sentence on water content:

> *[Ice softness] increases with liquid water fractions up to 0.01 (Duval, 1977; Lliboutry and Duval, 1985; Cuffey and Paterson, 2010, p. 65–66), an arbitrary threshold above which new ice deformation measurements are critically needed (Kleiner et al., 2015).*

The uncertainty to unknown rheology of water-rich temperate ice was also mentioned in the conclusions:

> *In the absence of ice deformation measurements, a constant rheology was used for temperate ice containing more than 1 % of liquid water.*

**p. 4, l. 19–20:** The sub-glacial hydrologic component should be described more (even

if exactly as in Bueler and van Pelt, 2015). Is basal water transported horizontally down the hydropotential gradient? This is usually a highly uncertain component of ice-sheet models, but can have a large effect on results through its influence on basal sliding, and basal frozen vs. thawed areas, which is relevant to section 4.4 regarding trimlines.

We refer to Bueler and van Pelt (2015) as their paper contain the most up-to-date description of PISM till effective pressure physics used in our simulations (Bueler and van Pelt, 2015, Eqs. 18, 23, and 24). However, subglacial water is not routed down the hydropotential gradient. We have clarified this:

*Effective pressure is related to the ice overburden stress and the modelled amount of subglacial water, using a formula derived from laboratory experiments with till extracted from the base of Ice Stream B in West Antarctica (Table 1; Tulaczyk et al., 2000; Bueler and van Pelt, 2015, Eqs 23 and 24). Basal meltwater is accumulated locally without transportation. When the till becomes saturated, additional meltwater assumed to drain off instantaneously outside the glacier margins, i.e. it is removed from the system in an accountable way.*

**Somewhat related:** Little information is given on the choices of basal sliding parameter values in Table 1. This could be discussed briefly. Presumably no inversion or optimization was performed for these values beforehand, and they do vary spatially. Are they appropriate for Alpine bedrock overall?

Although we refrain from repeating parameter values given in Table 1 in the main text, the following text was added in the methods:

*[a constant basal friction angle] corresponding to the average of available measurements (Cuffey and Paterson, 2010, p. 268). [...] Other parameters*

*(Table 1) follow simulations of the Greenland ice sheet (Aschwanden et al., 2013), or benchmarks when other data is missing (Bueler and van Pelt, 2015).*

Inversion of specific basal sliding parameters for past Alpine glaciers is difficult because the altitude and age of maximum ice surface elevation is discussed (cf. introduction and discussion on trimlines), and also depends on the even more uncertain regional climate history. Therefore, basal sliding was also mentioned as one of the major sources of uncertainty in the conclusions.

*The till deformation model used here does not hold for sliding over bedrock surfaces. On the other hand, the constant friction angle used is representative of wet till but weaker basal conditions may have applied over saturated lake sediments where they occured.*

**p. 7, l. 7:** Are there any data to support this atmospheric lapse rate value ($6\,\mathrm{K\,km^{-1}}$), and do other values have the potential to significantly affect ice temperatures? In particular, could they change the basal areas of frozen/unfrozen ice and so the comparisons with trimlines in section 4.4?

In the European Alps, monthly temperature lapse rates vary between approximately $4\,\mathrm{K\,km^{-1}}$ in winter and $7\,\mathrm{K\,km^{-1}}$ in summer, and annual temperature lapse rates vary spatially between 5.4 and $5.8\,\mathrm{K\,km^{-1}}$ (Rolland, 2003). A reference to the study by Rolland (2003) was added in Table 1, and as we have now explicited, our constant value of $6\,\mathrm{K\,km^{-1}}$ is thus:

*slightly above average but more representative of summer months when surface melt occurs (Rolland, 2003, Fig. 3).*

Although no sensitivity tests were conducted, we argue here that the effect of atmospheric temperature lapse rate variations on ice temperature is negligible. First, seasonal variations do not penetrate ice or bedrock below a few metres. Second, even above 2 km of ice, spatial variations of $0.4\,K\,km^{-1}$ would translate into surface temperature variations of only $0.8\,K$. But during the Last Glacial Maximum, ice surface temperatures in the trimline region are typically between 15 and 20 K below freezing. Thus the effect on the temperature gradient would be small. Actually, the effect would even nearly disappear near the glacier base, where the temperature gradient is much steeper and primarily controlled by geothermal heat flux and shear heating.

**Climate forcing**

The method of spatially uniform shifts to modern climate forcing is common in paleo-modeling of large ice sheets, and in my opinion is acceptable as a starting point in this work, with coupling to regional climate models (RCMs) left to follow-on work. There are good discussions on possible shortcomings of this method, for instance as a cause of anomalous east-west marginal ice extents at LGM (pg. 11, line 6-8). However, I suggest changing the sentence on pg. 9, line 5, which mentions some RCMs applied to LGM Europe, but also says "...over the Alps during the last glacial cycle, of which little is known apart from the LGM". There are several other RCM modeling studies over Europe during the last 120 kyrs, e.g., for MIS Stage 3, Kjellstrom et al., Boreas, 2010; Barron and Pollard, Quat. Res., 2002; Alfano et al., Quat. Res., 2002; and for 6 ka, Strandberg et al., Clim. Past, 2014. Perhaps there is little useful material there for the Alps, but such papers exist.

Thank you. We did not know about references on MIS 3 and have developed our sentence to better reflect the current state of knowledge on palaeo-precipitation:

> *Palaeoclimate proxies indicate slightly reduced LGM precipitation in western Europe with anomalies diminishing eastwards (Wu et al., 2007). Re-*
*gional circulation models indicate generally dryer conditions during MIS 3
(Barron and Pollard, 2002; Kjellström et al., 2010) but more precipitation
south of the Alps during MIS 2 (Strandberg et al., 2011; Ludwig et al., 2016).*

The past climate variations are prescribed following 3 quite distal core records, and
the most distal (EPICA) is chosen as yielding the best fit to Alpine glacial evidence.
The basis for preferring EPICA seems reasonable (matching some higher-frequency
amplitudes of ice variability, section 3.3). However, this agreement is in a sense co-
incidental, in that there is no direct meteorological link between Antarctic and Alpine
regional climate variations. Are there any proximal proxy records of Alpine climate at
all, perhaps lacustrian varves, that could be used to assess the EPICA-based shifts in
air temperatures and precipitation, even over limited periods of the last 120 kyrs?

A short review of available proximal proxy records was added here:

*Only few regional proxy records exist that extend over periods when the
Alps were glaciated (Heiri et al., 2014). These include lake sediment
records in north and west of the Alps (de Beaulieu and Reille, 1992; Wohl-
farth et al., 2008; Duprat-Oualid et al., 2017, e.g.,), and cave speleothems in
the Eastern (e.g., Spötl and Mangini, 2002; Boch et al., 2011) and Western
(Luetscher et al., 2015) Alps. Due to the scarcity of vegetation north of the
Alps during glacial periods, varying sources for moisture advection, and the
limited duration of the records, quantitative palaeoclimatic interpretation will
require combining multiple proxies in space and time, and comparing them
against regional circulation model output (Heiri et al., 2014).*

In the review paper by Heiri et al. (2014), the latter has been specifically identified as a
neeeded future development, but to our knowledge, no quantitative reconstruction has

been made available since then. The following word of caution was also added in the conclusions:

> *This suprising result is likely coincidental as there is no direct link between European and Antarctic climate. This highlights the need for more quantitative reconstructions of European palaeoclimate.*

A PDD scheme is used based only on seasonal air temperatures. van de Berg et al. (Nature Geosc., 2011) showed that for long-term variations including the Eemian, orbital changes in insolation are important and should be considered explicitly. This could be particularly relevant here, because the EPICA core does not reflect changes in insolation over the Alps. In further work, an insolation-change term (summer, local) could be combined with EPICA in the climate temperature paleo-forcing.

We agree. This is one of many possible improvements that could be tested upon the climate forcing used here. In the conclusions we now advocate for:

> *a more realistic climate forcing based on regional circulation model output or including the effect of long-term term changes in incoming solar radiation.*

**Trimlines**

The point is well taken that trimlines do not necessarily indicate past ice surface elevations, but the upper limit of temperate ice with cold ice above (pg. 18, line 5 to pg. 19, line 4). It is an important point, because model LGM ice surfaces are far above most trimline elevations (as noted and in Fig. 6a,b). The pertinent results are shown

in Fig. 6c, and I agree there is good support for the cold vs. temperate (basal) ice hypothesis.

Thank you for emphasizing this point.

It might help general readers to spell out the interpretation even more in the text. That is, as I understand it, the observed trimlines should coincide with the boundaries between model areas of frozen vs. temperate beds, so the dots in Fig. 6c should all lie on the borders between the hatched and white areas. The sentence on p. 19, l. 20-21, is confusing in this regard. Incidentally, it would also help to add the word "basal" to the last sentence of the Fig. 6c caption: "...experienced temperate basal ice for ...".

We realise that Fig. 6 is confusing. We had chosen the 1 ka limit because temperate basal ice tends to occur above the trimlines during short periods of warmer climate. To avoid confusion, Fig. 6c has been simplified to show the basal thermal boundary at the age of maximum ice thickness. The last sentence in the caption now reads:

*Hatches mark the LGM cold-based ares (basal temperature above $1 \times 10^{-3}$ K below freezing at the age of maximum ice thickness).*

Unfortunately this also results in a worse fit to trimline locations, as was noted in the main text:

*In the upper Rhone Valley, observed trimlines are often located near the LGM cold-temperate basal thermal transition, or in cold-based areas (Fig. 6c).*

One reason for the remaining discrepancies in Fig. 6c could be temporal variations in the model boundaries, that are aggregated in time by the "< 1 kyr" criterion for the

hatching and the grouping of all trimline data. To go into this in more detail, in principle Fig. 6c could be expanded to show the model basal frozen-temperate boundaries at particular times (21.5, 22.5, etc, ka), with only the dots for each time period superimposed. But that may not be worth it unless there are large temporal variations in the model boundaries.

Fig. 6c has been simplified to show the basal thermal boundary at the age of maximum ice thickness. Although this results in a time-transgressive picture, it means that the frozen areas indicated on the new figure are contemporaneous with the plotted ages and surface topography contours, and data shown on panels a and b.

Although the model output allows to study the migration of the basal thermal boundary over time into more detail, the basal velocity, which also controls erosion, should probably be considered as well. We leave this for future studies. The following sentence was added in the main text:

> *The remaining discrepancies may relate to temporal migrations of the basal thermal boundary, an absence of sliding in warm-based areas [...]*

A slight concern is that the majority of the trimline data seems to be orange dots i.e., older that 27 ka in the timescale of Fig. 6c. The period for the model hatching extends back only to 29 ka. Hopefully, most of the orange-dotted data are within that period and are not older than 29 ka(?).

In fact, much of the mountainous areas reach maximum ice thickness during early MIS 2 when ice is colder and stiffer. This is also visible on Fig. 5. Some areas even reach maximum ice thickness during MIS 4 when the bedrock is slightly less depressed. There is certainly very much scope to discuss the age of the trimlines, which may differ significantly from that of the maximum ice extent on the lowland, but we are not aware of any data that could be used for validation here. However, frozen

areas plotted on Fig. 6c are now consistent with the modelled ages used in panels a and c.

The text could briefly mention (and hopefully rule out) the issue of very fine-scale to-pographic features on which the trimlines are located, not resolved by the 1-km model topography. If data sites are on small-scale highs or lows significantly different from their km-scale surroundings, that could contribute to the discrepancies in Fig. 6c.

This is a valid point which we can unfortunately not rule out. Despite PISM's enthalpy scheme, which ensures a seamless transition between cold and temperate ice physics, resolving the basal thermal boundary on the valley sides is delicate, because it implies resolving a steep (vertical) enthalpy gradient over a steep bedrock slope. This requires both high vertical and high horizontal grid resolutions. This limitation was mentioned in the section on trimlines:

> [The remaining discrepancies may relate to] levelling of small-scale topo-graphic features in the 1 km horizontal grid. They call for more detailed comparisons [...]

**Technical comments**

**p. 3, l. 2:** Perhaps change "lead" to "led", "to which" to "to what".

Done.

**Fig. 1 caption, l. 4:** Perhaps change "estimated" to "estimate", or "estimated of" to "estimated".

Done. We thank you very much again for the time and effort you put into our manuscript.

[Figure]

**References**

Aschwanden, A., Bueler, E., Khroulev, C., and Blatter, H.: An enthalpy formulation for glaciers and ice sheets, J. Glaciol., 58, 441–457, doi:10.3189/2012JoG11J088, 2012.

Aschwanden, A., Aðalgeirsdóttir, G., and Khroulev, C.: Hindcasting to measure ice sheet model sensitivity to initial states, The Cryosphere, 7, 1083–1093, doi:10.5194/tc-7-1083-2013, 2013.

Barron, E. and Pollard, D.: High-Resolution Climate Simulations of Oxygen Isotope Stage 3 in Europe, Quaternary Res., 58, 296–309, doi:10.1006/qres.2002.2374, 2002.

Blatter, H. and Greve, R.: Comparison and verification of enthalpy schemes for polythermal glaciers and ice sheets with a one-dimensional model, Polar Sci., 9, 196–207, doi:10.1016/j.polar.2015.04.001, 2015.

Boch, R., Cheng, H., Spötl, C., Edwards, R. L., Wang, X., and Häuselmann, P.: NALPS: a precisely dated European climate record 120–60 ka, Clim. Past, 7, 1247–1259, doi:10.5194/cp-7-1247-2011, 2011.

Bradford, J. H. and Harper, J. T.: Wave field migration as a tool for estimating spatially continuous radar velocity and water content in glaciers, Geophys. Res. Lett., 32, doi:10.1029/2004gl021770, 2005.

Bradford, J. H., Nichols, J., Mikesell, T. D., and Harper, J. T.: Continuous profiles of electromagnetic wave velocity and water content in glaciers: an example from Bench Glacier, Alaska, USA, Ann. Glaciol., 50, 1–9, doi:10.3189/172756409789097540, 2009.

Bueler, E. and van Pelt, W.: Mass-conserving subglacial hydrology in the Parallel Ice Sheet Model version 0.6, Geosci. Model Dev., 8, 1613–1635, doi:10.5194/gmd-8-1613-2015, 2015.

Cuffey, K. M. and Paterson, W. S. B.: The physics of glaciers, Elsevier, Amsterdam, 2010.

de Beaulieu, J.-L. and Reille, M.: The last climatic cycle at La Grande Pile (Vosges, France) a new pollen profile, Quaternary Sci. Rev., 11, 431–438, doi:10.1016/0277-3791(92)90025-4, 1992.

Duprat-Oualid, F., Rius, D., Bégeot, C., Magny, M., Millet, L., Wulf, S., and Appelt, O.: Vegetation response to abrupt climate changes in Western Europe from 45 to 14.7k cal a BP: the Bergsee lacustrine record (Black Forest, Germany), J. Quaternary Sci., 32, 1008–1021, doi:10.1002/jqs.2972, 2017.

Duval, P.: The role of the water content on the creep rate of polycrystalline ice, in: Grenoble Symposium, 1975, isotopes and impurities in snow and ice, vol. 118 of *IAHS Publ.*, pp. 29–

33, 1977.

Greve, R.: A continuum-mechanical formulation for shallow polythermal ice sheets, Philos. T. R. Soc. A, 355, 921–974, doi:10.1098/rsta.1997.0050, 1997.

Heiri, O., Koinig, K. A., Spötl, C., Barrett, S., Brauer, A., Drescher-Schneider, R., Gaar, D., Ivy-Ochs, S., Kerschner, H., Luetscher, M., Moran, A., Nicolussi, K., Preusser, F., Schmidt, R., Schoeneich, P., Schwörer, C., Sprafke, T., Terhorst, B., and Tinner, W.: Palaeoclimate records 60–8 ka in the Austrian and Swiss Alps and their forelands, Quaternary Sci. Rev., 106, 186–205, doi:10.1016/j.quascirev.2014.05.021, 2014.

Kjellström, E., Brandefelt, J., Näslund, J.-O., Smith, B., Strandberg, G., Voelker, A. H. L., and Wohlfarth, B.: Simulated climate conditions in Europe during the Marine Isotope Stage 3 stadial, Boreas, 39, 436–456, doi:10.1111/j.1502-3885.2010.00143.x, 2010.

Kleiner, T. and Humbert, A.: Numerical simulations of major ice streams in Western Dronning Maud Land, Antarctica, under wet and dry basal conditions, J. Glaciol., 60, 215–232, doi:10.3189/2014jog13j006, 2014.

Kleiner, T., Rückamp, M., Bondzio, J. H., and Humbert, A.: Enthalpy benchmark experiments for numerical ice sheet models, The Cryosphere, 9, 217–228, doi:10.5194/tc-9-217-2015, 2015.

Lliboutry, L. A. and Duval, P.: Various isotropic and anisotropic ices found in glaciers and polar ice caps and their corresponding rheologies, Ann. Geophys., 3, 207–224, 1985.

Ludwig, P., Schaffernicht, E. J., Shao, Y., and Pinto, J. G.: Regional atmospheric circulation over Europe during the Last Glacial Maximum and its links to precipitation, J. Geophys. Res. Atmos., 121, 2130–2145, doi:10.1002/2015jd024444, 2016.

Luetscher, M., Boch, R., Sodemann, H., Spötl, C., Cheng, H., Edwards, R. L., Frisia, S., Hof, F., and Müller, W.: North Atlantic storm track changes during the Last Glacial Maximum recorded by Alpine speleothems, Nature Communications, 6, 6344, doi:10.1038/ncomms7344, 2015.

Murray, T., Stuart, G. W., Fry, M., Gamble, N. H., and Crabtree, M. D.: Englacial water distribution in a temperate glacier from surface and borehole radar velocity analysis, J. Glaciol., 46, 389–398, doi:10.3189/172756500781833188, 2000.

Murray, T., Booth, A., and Rippin, D. M.: Water-content of glacier-ice: limitations on estimates from velocity analysis of surface ground-penetrating radar surveys, J. Environ. Eng. Geoph., 12, 87–99, 2007.

Rolland, C.: Spatial and Seasonal Variations of Air Temperature Lapse Rates in Alpine Re-
gions, J. Climate, 16, 1032–1046, doi:10.1175/1520-0442(2003)016<1032:sasvoa>2.0.co;2, 2003.

Rückamp, M., Blindow, N., Suckro, S., Braun, M., and Humbert, A.: Dynamics of the ice cap on King George Island, Antarctica: field measurements and numerical simulations, Ann. Glaciol., 51, 80–90, doi:10.3189/172756410791392817, 2010.

Seroussi, H., Morlighem, M., Rignot, E., Khazendar, A., Larour, E., and Mouginot, J.: Dependence of century-scale projections of the Greenland ice sheet on its thermal regime, J. Glaciol., 59, 1024–1034, doi:10.3189/2013jog13j054, 2013.

Spötl, C. and Mangini, A.: Stalagmite from the Austrian Alps reveals Dansgaard–Oeschger events during isotope stage 3:: Implications for the absolute chronology of Greenland ice cores, Earth Planet. Sc. Lett., 203, 507–518, doi:10.1016/S0012-821X(02)00837-3, 2002.

Strandberg, G., Brandefelt, J., Kjellstrom, E., and Smith, B.: High-resolution regional simulation of last glacial maximum climate in Europe, Tellus A, 63, 107–125, doi:10.1111/j.1600-0870.2010.00485.x, 2011.

Tulaczyk, S., Kamb, W. B., and Engelhardt, H. F.: Basal mechanics of Ice Stream B, west Antarctica: 1. Till mechanics, J. Geophys. Res., 105, 463, doi:10.1029/1999jb900329, 2000.

Wohlfarth, B., Veres, D., Ampel, L., Lacourse, T., Blaauw, M., Preusser, F., Andrieu-Ponel, V., Kéravis, D., Lallier-Vergès, E., Björck, S., Davies, S. M., de Beaulieu, J.-L., Risberg, J., Hormes, A., Kasper, H. U., Possnert, G., Reille, M., Thouveny, N., and Zander, A.: Rapid ecosystem response to abrupt climate changes during the last glacial period in western Europe, 40–16 ka, Geology, 36, 407, doi:10.1130/G24600A.1, 2008.

Wu, H. B., Guiot, J. L., Brewer, S., and Guo, Z. T.: Climatic changes in Eurasia and Africa at the last glacial maximum and mid-Holocene: reconstruction from pollen data using inverse vegetation modelling, Clim. Dynam., 29, 211–229, 2007.

---

## Author Comment (AC3) · 23 Jul 2018

Dear Anonymous Referee #2,

Thank you very much for your detailed review of our manuscript.

**General comments**

This manuscript describes a model study with the Parallel Ice Sheet Model applied to the last glacier cycle in the Alps. The climate forcing is derived from present day climate of WorldClim and the ERA-Interim reanalysis and time-dependent temperature offsets derived from the Greenland Ice core (GRIP), from Antarctic ice core (EPICA)

[Figure]

and Marine sediment core from the Iberian margin (MD01-2444). The study is split in two, in the first part analysis of six model simulations (with and without precipitation scaling) made on a 2 km resolution grid is presented and concluded that out of the three climate forcing records used, the EPICA record gives the most realistic ice volume history during MIS 4 and 2. The second part analyses simulation made with the EPICA forcing record on a 1km grid. The authors draw conclusions about the ice cover, ice flow pattern, ice thickness, LGM ice extent, which in their model is a transient stage with varying timing across their model domain due to glacier dynamics.

Thank you very much for this fully accurate summary.

The manuscript is well organized and clearly written, the missing thing in this study is a discussion of and preferably a sensitivity study of the ice dynamic-model assumptions made. Is the sliding of the ice age ice sheet realistically modelled with pseudo-plastic assumption using the Shallow Shelf approximation? How sensitive are the results to the selected model parameters? It is briefly mentioned once on page 19, line 8, but a thorough analysis of the model sensitivity would strengthen the paper.

Thank you. We agree that a sensitivity study to ice physics, in particular basal sliding, is one of many improvements needed upon our results. However, for the kind of exercise presented here, the sensitivity to palaeo-climate history, which is virtually unknown and has many degrees of freedom, far outweight the model sensitivity to ice physics. From the perspective of glacial geology, a 5 K change in air temperature is more influential than an order-of-magnitude increase in ice velocities (perhaps with the exception of our conclusions on trimlines, see below). In its current state, our manuscript studies the model sensitivity to the primary source of uncertainty, which is climate.

Because the simulations presented here depend on computing resources only available through a peer-review application process, we are currently unable to perform an additional sensitivity study on basal sliding for the entire Alps. However, a regional study on the model sentivity to basal sliding is actually the topic for a follow-up study

currently under preparation for the Rhine Glacier (Imhof et al., 2017), which has a relatively isolated catchment for its size as compared to other Alpine Glaciers.

Meanwhile, the following text was added in the section on trimlines, which is certainly the most sensitive aspect of our results to basal sliding:

*The remaining discrepancies may relate to temporal migrations of the basal thermal boundary, an absence of sliding in warm-based areas, and levelling of small-scale topographic features in the 1 km horizontal grid. They call for more detailed comparisons spanning the entire Alpine range and specific sensitivity studies to relevant basal sliding and ice rheological parameters, and to the uncertain subglacial topography.*

In addition, basal sliding and other sources of uncertainties in ice physics where also mentioned in the conclusions:

*However, these results are limited by uncertainties on ice physics. The till deformation model used here does not hold for sliding over bedrock surfaces. On the other hand, the constant friction angle used is representative of wet till but weaker basal conditions may have applied over saturated lake sediments where they occured. In the absence of ice deformation measuremets, a constant rheology was used for temperature ice containing more than 1 % of liquid water.*

**Specific comments**

I find missing something that indicates that the times are before present, as in line 5, line 8 and line 16 on page 1 – and elsewhere in the paper it is written (120-0 ka) should you add before present or BP to indicate the time interval?

We have now defined this better in the introduction:

*During the last 800 ka (thousand years before present)*

Could the reason for too large ice volumes when using the GRIP record, as mentioned in lines 13-15 on page 9 be due to Arctic Amplification? Could that have an effect then as it has now? This is also mentioned in line 11 on page 10.

These passages refer to the Younger Dryas and Dansgaard-Oeschger events. One must keep in mind that a linear scaling was applied to the GRIP record, so that it is already "de-amplified" before being used as temperature forcing. But still, Arctic amplification could be an explanation if it was stronger then than for changes that lead the LGM climate. Although we lack expertise on this topic, we could not find literature supporting this idea. Instead, current theories for the Younger Dryas and Dansgaard-Oeschger events mostly involve large amounts of freshwater discharge into the North Atlantic implying a more regional mechanism than Arctic amplification. To avoid restrictive interpretations, we thus decided not to mention Arctic amplification as a possible explanation.

**Minor comments**

**p. 1, l. 24:** suggest: "have extended well outside their current margins".

Thank you. Done.

**p. 2, l. 22:** could you add a reference and a timing for LGM?

Done.

**p. 3, l. 1:** suggest to replace "thus" with "still".

Done.

**p. 3, l. 3:** add s to responses.

Done.

**p. 3, l. 18:** something missing in the sentence, suggest "formulation" after "creep".

Done.

**p. 4, l. 4:** suggest to replace "field" with "sheet".

Done.

**p. 5, Fig. 1:** c) can you add a scale and maybe indication with a box in b) where this extract is from? "from the estimate" (not plural in line 4 of caption), Line 6 (PDD) is not acronym for surface mass balance, some more explanation is needed here, indicate also, like in the other figures that h) is January and i) is July figures.

We have added a scale, marked the inset location, and clarified the text.

**p. 6, l. 6:** suggest to replace "of" with "with".

Done.

**p. 6, l. 14:** suggest "The climate forcing driving the ice sheet simulations consist spatially of a present-day monthly mean climatology..."

Done.

**p. 6, l. 15:** suggest to delete "s" on corrections.

Done.

**p. 6, l. 19:** add "mean" after monthly.

Done.

**p. 6, l. 22:** note, if true (clarify in Figure caption, see comment above) then the reference should be to Fig. 1 i) for summer precipitation.

This is true, done.

**p. 6, l. 27:** replace "is" with "are".

Done.

**p. 6, footnote:** add a reference for the correct formula for the rigidity and clarify (add something like, the consequence of this error is that the simulations effectively use....) Delete "in" before "a small" and add a quantification of the small change in the length scale – how small, is it a few percentage?

The correct formula can be found in the study by Love (1906, p. 443). The lenght scale of bedrock deformation can be computed as $\alpha = \sqrt[4]{\frac{4D}{(\rho_m - \rho_l)g}}$ (Walcott, 1970), thus the error in lenght scale can be expressed as $\sqrt[4]{1 + \nu} - 1 = 5.7\%$. The footnote was clarified and references were added.

**p. 7, l. 2:** "shipped with WorldClim" is not clear, please edit, also suggest not to use b for topography, s surf and s bed , or h would be better.

We clarified the sentence. However, we would like to keep the current notation to remain consistent with previous papers (Seguinot, 2014; Seguinot et al., 2014).

**p. 7, l. 19:** add "the" before "oxygen".

Done.

**p. 7, l. 24:** "and within a rectangular region .." is not clear, edit this text.

The sentence was reorganised.

**p. 8:** add "acceleration of" before gravity, add a reference for the ideal gas constant?

Done.

**p. 9, l. 14–15:** see comment above, could this be an example of Arctic amplification?

Please refer to my comment above.

**p. 9, l. 19:** add "a" before "very" and suggest to turn the sentence around, the EPICA simulations are in a good agreement with the data.

Done.

**p. 9, l. 20:** followed by first a retreat and then a standstill? replace "blue" with "red".

Done.

**p. 9, l. 21:** suggest : The simulations forced by the GRIP palaeo-temperature forcing yield ... (blue curves)".

Done.

**p. 10, l. 2:** suggest to add "followed by a rapid retreat".

Done.

**p. 10, l. 3:** why state "two or three" in intro the suggestion is either 4 major or 15, is there a reference for 2 or 3?

The numbers 4 and 15 relate to the total (minimum) number of Pleistocene glaciations. However, evidence for pre-LGM glaciations during the last glacial cycle is debated and very limited (see later part of the introduction). We added the word "Pleistocene" in the introduction and clarified the concerned sentence about model results.

**p. 10, l. 10:** suggest to replace "lower" with "smaller".

Done.

**p. 10, l. 11:** is this due to Arctic Amplification?

Please refer to my comment above.

**p. 10, l. 12:** suggest to replace "least" with "the smallest".

Done.

**p. 11, Fig. 3 caption:** what does "cumulative" mean here? Do you mean maximum in each location? clarify what the black line indicates. No solid red line is visible in figures. What is meant with "reasonable", maybe replace with "realistic"? suggest to replace "cover" with "extent".

We replaced "cumulative extent of modelled ice cover" with "modelled maximum extent", "reasonable" with "realistic", and "red line" with "black line", which is what we meant.

[Figure]

**p. 11, l. 2:** suggest to replace "cumulative" with "maximum extent in each area" or something similar.

As for the caption, we replaced with "modelled maximum extent".

**p. 11, l. 1–3:** how sensitive is the model to different parameters in the applied sliding formulation?

Please refer to my comment above.

**p. 11, l. 4–5:** "outside this benchmark" clarify if you mean spatially or temporally.

Done. We mean spatially.

**p. 11, l. 6–11:** is this text better fitted in a discussion section?

We have included this discussion here as we think that understanding the simiarities and differences in the spatial patterns of glaciation between the different palaeo-temperature records used is necessary to justifying the choice of the EPICA forcing used in the next section. We think that this non-standard outline is the best way to present all results in a single manuscript.

**p. 12, l. 2:** suggest to replace "higher" with "larger".

Done.

**p. 12, l. 4:** do you mean to refer to Fig. 3?

No, but the reference to Fig. 2 was misplaced. This has been corrected.

**p. 12, l. 29–32:** does this text fit better in method section?

We agree that this text does not really fit in a results and discussion section. However, this information is specific to Sect. 4. We feel that moving these and the corresponding sentences in Sect. 3 to the methods section would increase the complexity of the manuscript.

**p. 13, Fig. 4 caption:** the 200 m surface contours are not clearly visible in the figure, can they be made sharper, or just skipped? Line 3, something like "are shown" is missing. Suggest to replace "background" with "bedrock". This is the first time "Natural Earth Data" is mentioned, should that be in the section on the data? Suggest to replace "Gray fields" with "shaded gray area" and "boundaries" with "timing".

The 200 m contours were thickened, the caption was reworked as suggested, and the reference to Natural Earth Data was moved to the caption of Fig. 1.

**p. 15, l. 8:** replace "was" with "is".

Done.

**p. 15, l. 25:** "occurred".

Done.

**p. 16, Fig. 5:** in figure the color bar is written to indicated maximum surface elevation, but in figure caption the maximum ice thickness, which is correct? The surface elevation contours are not clearly visible. The dark orange color in the figure, that covers the central part of the ice sheet is not (?) in the bar on the left (or is it before 27 ka BP?).

The map shows maximum thickness. The colourbar label was corrected. Surface elevation contours were thickened. Indeed, in much of the mountain area maximum thickness is reached before 27 ka. This is now clarified in the figure caption.

**p. 17, l. 30:** suggest to replace "have" with "could".

Done.

**p. 18, Fig. 6:** the 200 m contours are not clearly visible.

The 200 m contours were thickened.

**p. 19, l. 7:** could you add what the modelled regional ice thickness is, for comparison?

Done.

**p. 20, Fig. 7:** add info about the gray areas indicating MIS 4 and 2 "Isolated patches indicate periodic surges from tributary glaciers" needs more explanation and it is not clear what is meant. Does the model simulate periodic surges?

We added the info about MIS. The wording "periodic surges" was badly chosen and replaced by a more cautious "episodic advances". These advances are likely due to thermodynamical feedbacks modulated by fluctuations in the climate forcing, but they were not explored in detail and perhaps depend on model resolution.

**p. 21, l. 21:** here is a reference for 2 or 3 glaciations (see comment above) but in the intro is only mentioned 4 or 15, suggest to change text to harmonize.

See my previous comment about last glacial cycle and older glaciations.

**p. 21, l. 27:** suggest to edit "the study consists of" or start for example like "In this study the model has been applied..."

Done.

**p. 21, l. 28:** how important is it that the model has been validated for the Cordilleran ice sheet? Will that support the choices of the sliding model applied in the Alps? I think that it would be valuable for this study to do a sensitivity runs for at least some of the model parameter choices.

The reference to the Cordilleran ice sheet was removed here. As mentioned in the later part of the conclusions, it is clear to us that additional sensitivity studies are needed. With this study, we hope to have highlighted areas of research were data-model comparison could be pursued further. However, this will require not only more model runs, but also a more systematic classication of geological data in digital databases.

**p. 22, l. 6:** add "records" after forcing.

We replaced "forcing" by "records".

**p. 22, l. 19:** why do you add "potentially" here? Isn't this a firm conclusion from you study?

The timing of the LGM extent depends on the actual climate history of the European Alps, which was almost certainly different and more complex than our climate forcing scenarii. Although our simulations show that even a spatially homogeneous forcing produces spatial variations in the timing of maximum extent, on can not exclude that a different climate history might counteract these variations.

**p. 22, l. 20:** suggest to replace "higher" with "larger".

Done.

**p. 22, l. 22:** same as above, why "potentially" here?

We removed the first "potentially". Nevertheless, the actual number and timing of

glacier advances onto the foreland depends on Alpine climate history, which may have been very different from the forcing used in our simulations.

**p. 22, l. 24:** suggest to replace "nevertheless" with "however" or edit the sentence.

Done.

**p. 22, l. 26–28:** the paper would be able to give stronger conclusions with sensitivity study, suggest to edit the sentence by replace "statements" with "limitations" or "drawbacks" and "last glacier cycle ice dynamics in the Alps" is not easy to read.

Done. The complicated sentence was rephrased.

**p. 23, l. 2:** replace "mode" with "more".

Done.

Figures are generally clear and well set up. The surface contours in Figs 4,5 and 6 is not clearly visible in my printout and could maybe become clearer?

Thank you. The 200 m contour levels were thickened on all figures. We thank you again very much for the time and effort you put into this very meticulous review.

**References**

Imhof, M., Jouvet, G., Seguinot, J., and Funk, M.: Modeled and reconstructed ice thickness of the Rhine Glacier during the Last Glacial Maximum, in: EGU General Assembly Conference Abstracts, vol. 19, p. 13681, 2017.

Love, A. E. H.: A treatise on the mathematical theory of elasticity, Cambridge University Press, 2 edn., 1906.

Seguinot, J.: Numerical modelling of the Cordilleran ice sheet, Ph.D. thesis, Stockholm University, http://urn.kb.se/resolve?urn=urn:nbn:se:su:diva-106815, 2014.

Seguinot, J., Khroulev, C., Rogozhina, I., Stroeven, A. P., and Zhang, Q.: The effect of climate forcing on numerical simulations of the Cordilleran ice sheet at the Last Glacial Maximum, The Cryosphere, 8, 1087–1103, doi:10.5194/tc-8-1087-2014, 2014.

Walcott, R. I.: Flexural rigidity, thickness, and viscosity of the lithosphere, J. Geophys. Res., 75, 3941–3954, doi:10.1029/JB075i020p03941, 1970.
* * *

---

## Author Comment (AC4) · 23 Jul 2018

Dear Anonymous Referee #3,

Thank you very much for your detailed review of our manuscript.

This is an excellent manuscript and I highly recommend publication after a few minor changes. The authors present a well thought out modelling experiment which they combine (albeit in a qualitative manner) with extensive palaeo-glaciological data and cumulative work. I can see this work being extended into more extensive and rigorous work (RCM forcing, ice dynamic sensitivity, quantitative fitting to geomorphological record etc), but this is an important leap forward.

[Figure]

Thank you very much for these supportive words!

**Abstract**

**p. 1, l. 2:** "pioneer" should be "pioneering".

Thank you. Corrected.

**p. 1, l. 16:** I think the finding that you get asynchronous glaciation extents with a uniform climate offset is due to glacier hypsometry and setting should be stated here. i.e. that the timing of maximum glaciation and recession isn't purely a function of climate. This finding needs to be highlighted better in the abstract.

Thank you for pointing this out. We have reworked the last sentence in the abstract:

*Finally, despite the uniform climate forcing, differences in glacier catchment hypsometry cause the Last Glacial Maximum advance to be modelled as a time-transgressive event, with some glaciers reaching their maximum as early as 27 ka, and others as late as 21 ka.*

**Introduction**

**p. 2, l. 28:** Ballantyne and Stone (2015) should be added to this list.

Thank you. This is relevant indeed. We have added the reference in the introduction and in the corresponding section of the discussion.

**p. 2, l. 34:** It should be stated that it could be a consequence of both glaioclimatic interactions and uncertainties in dating methods.

We have added "or both".

**p. 3, l. 2:** These points serve the literature well to highlight gaps for future research. However, I would argue that you do not get very far here on 1 and 5 and do not completely solve the other 3 points. Your text reflects these shortcomings very well, for which you should be applauded. Though I think at this stage of the manuscript, your statement of intent, you should state that you do not claim to solve these questions, but rather push forward on all of them using your new approach of ice sheet modelling.

Thank you. In order to clarify our statement of intent, we added the following text:

*Additional geological research will be needed to complete our knowledge. But here, we intend to explore these open questions from a new angle and [use the Parallel Ice Sheet Model ...].*

**Section 2.6**

**p. 7:** The spatial distribution of your modern climate variables (precip, temp) will be massively influenced by elevation. Though there is a lapse rate, does this pattern of high precip and low temp over mountains remain throughout the simulation despite ice surface topography, and if so, how does this influence your results?

The temperature lapse rate corrections dampen local high temperatures in ice-filled valleys. The precipitation pattern, on the other hand, remains the same throughout the simulation, and this results in a higher accumulation above the location of present-day mountains even as they are buried by ice in the model.

Although we expect that small-scale variations in surface mass balance are smoothed out by ice flow, the presence of an ice sheet may have redistributed precipitation on

a larger scale, for instance from the inner ranges to the foothills, or inequally between north and south of the Alps, etc. However, global precipitation reductions appear to have little effect on the average ice thickness (Fig. 2b, compare dark and light curves). Nevertheless, the new text shortly mentions this caveat:

*Besides [circulation changes], changes in the ice surface topography may have redistributed orographic precipitation.*

Our conclusions also highlight the need for a more realistic forcing.

*Using [...] a more realistic climate forcing based on regional circulation model output [...] future modelling studies will certainly be able to quantify uncertainties associated with some of the above limitations.*

**Section 3**

**p. 7, l. 14:** You keep mentioning the number of processors. I find this information slightly irrelevant, and it will soon become outdated as processing speed and models increase (GPUs for example). The only way it will serve the community is if there is a full description of the computer set up. For example, it could be that the simulations took 4 days on 144 processors, but the processors were slow. Suggest removing these references.

We removed mentions of computing times and numbers of processors.

**p. 10, l. 14:** This is a more general point. You eventually choose the EPICA record, and for justifiable reasons based on comparison to reconstructed ice extent and timing. However, this is likely coincidence. EPICA is likely a complex record containing global

and local antarctic influences upon climate. The "real" climate over the alps during glaciation is like decoupled from that of Antarctica to an extent. Therefore, the match you find is not an inference about climate, as different combinations of offsets may have made the same result (smoothed GRIP to remove some of the D-O scale noise?). You should make this explicit somewhere in the manuscript.

We agree. The following text was added in the conclusions:

*This suprising result is likely coincidental as there is no direct link between European and Antarctic climate. This highlights the need for more quantitative reconstructions of European palaeoclimate.*

**p. 11, l. 2:** First sentence needs reconsidering as it is slightly broken in its current form. Perhaps "Figure 3a shows the cumulative extent of glaciated area during MIS2".

We reworked the first sentence. It now reads:

*During MIS 2, all six simulations yield comparative modelled maximum extents (Fig. 3a).*

**p. 11, l. 10:** Is it possible some ice is missing from the geological reconstruction in some instances? I guess some outlets are well constrained, whilst others areas could be "filled in" by this modelling experiment.

Yes, this is possible. Parts of the reconstructed LGM ice limits are uncertain due to the lack of moraines or dated evidence. However, without very detailed knowledge about palaeoclimate, and tight constraints on ice physics, the model won't replace observations. Actually, we are very confident that the model-data discrepancies in ice extent discussed in this passage are beyond uncertainties.

**Section 4**

**p. 14, l. 15–31:** I find this description of sites and timing of glacier extent compared to dates difficult to follow. I suggest a new figure to convey this important comparison: Have the reconstructed and modelled ice extents at key times for each of the mentioned glaciers on several smaller maps, including geochronological constraints.

This would be very useful but to our knowledge, no geological data have been made available that would allow such a comparison. The review by Wirsig et al. (2016, Fig. 5) which we refer to is the most up-to date compilation of LGM ages in and around the Alps, and deglacial outlines like there exists for the Eurasian and Laurentide ice sheets are unfortunately not available.

**p. 17, l. 17:** This finding is important and should be highlighted better in abstract and conclusion.

This finding was better highlighted in the abstract (see our previous comment). However, we feel it is already well expressed by the following bullet point in the conclusion:

> *The LGM (maximum) extent was a transient stage in which glaciers were out of balance with the contemporary climate. Its timing potentially varied across the range due to inherent glacier dynamics (Sect. 4.3).*

**p. 17, l. 22:** Is the model recreating possible surging? Or would this be an over-interpretation given the uncertainty in climate and physics. Seems to fit with the en-thalpy model of Benn and others for surging. As reads, it suggests that these areas were possible palaeo-surges, please clarify.

Our simulations reproduce a thermodynamical instability akin to a surge. The internal energy (so-called enthalpy) conservation scheme used in PISM allows for variations in

ice temperature and liquid water content (Aschwanden et al., 2012), which may result in thermodynamical oscillations (Pelt and Oerlemans, 2012; Feldmann and Levermann, 2017).

Unlike periodic surges (as described by, e.g., Sevestre and Benn, 2015), we interpret the model results as triggered by an initial perturbation of mass (and energy) caused by rapid climate cooling and inefficient mass transfer to lower elevations. We have rephrase the following sentence:

> *Several Alpine lobes surge and overshoot their balanced extent before thin-*
> *ning and receding towards the mountains as they warm towards thermody-*
> *namical equilibrium.*

**p. 19, l. 9–12:** On trimlines: I think you justify well why you haven't yet modelled the sensitivity to your trimline result - previous work backs this up. But, I think you should consider the following: Adding a plus/minus to your results to reflect the uncertainty. The importance of resolution - many trimlines will be below the resolution of your model, so perhaps aren't resolved enough for model-data comparison. They may have acted to deflect ice flow around mountain peaks for example. This resolution caveat should be mentioned. I would be surprised if all trimlines are subglacial transitions as this paper suggests - perhaps you need to directly challenge the geochronological community to find better trimline constraints (sub/supra) as a statement in this paper. Your mean value of 861 m is unrepresentative of your sample. Your sample is highly skewed, so a modal value (1050 m ish from Fig 6) is more appropriate. A similar finding with a similar approach has already been found for the British Isles, with the added constraint of GIA observations. I suggest referencing Kuchar et al. 2011 for this reason.

Thank you. This is a valid point. However we found an error in our figure: the data plotted on the histogram (Fig. 3b) were different from those in the scatter plot (Fig. 3a).

These data were raw interpolated ice thicknesses, i.e. they missed a correction for altitude shifts between our initial basal topography and the observed trimline elevations.

The corrected histogram is closer to symmetry. A slight skewness remains, but this is not really a result of our simulations. In fact the distribution of observations itself is skewed, with more trimlines observed at higher elevations, and there is a tendency for the differences between the modelled ice surface and trimlines to increase with altitude.

The modal value (now ca. 950 m) corresponds to a cluster of trimline locations around 2850 m in the Aletsch Glacier (south of Jungfrau) region where trimlines are well visible in the landscape. We prefer to stick with the mean because the number of trimlines mapped is not necessarily representative of their lenght, and may depend on other factors such as bedrock lithology and post-glacial erosion.

We now use the standard deviation (197 m) as a measure of variance, but not uncertainty, which would require more sensitivity tests. We also mention resolution as one of several caveats in the text:

> *The remaining discrepancies may relate to temporal migrations of the basal thermal boundary, an absence of sliding in warm-based areas, and levelling of small-scale topographic features in the 1 km horizontal grid.*

We added a reference to the study by Kuchar et al. (2012) although this approach is not doable for the Alps.

> *Unfortunately, validation through the bedrock uplift rate (cf. Kuchar et al., 2012) is not doable in the Alps due its lower values, active tectonics and uncertainties on geological properties (cf. Mey et al., 2016).*

A philosophical but important point is that your discussion throughout is written from the standpoint of the geochronological/geomorphological data and reconstructions as

being "truth". It should consider somewhere that perhaps data is missing as it is hard-won in the places it exists and interpretations may be slightly wrong. The model is also not the "truth" and there is probably a blurred line in-between upon which we can proceed.

Thank you. Indeed the quality and availability of geomorphological and geochronological data is very variable across the Alps, and long-standing interpretations are sometimes proven wrong (e.g., Monegato et al., 2007). Also, parts of the Alpine glacial geology community will certainly disagree with your comment and find our challenging of the trimline assumption quite provocative, given that it has been around for at least a century (Penck and Brückner, 1909).

**p. 21, l. 11–19:** Be really clear here that these are modelled, and perhaps not geologically recorded, advances of the ice sheet. If there is no data, it might be correct, might be just a modelled result.

Actually we meant geologically recorded asymmetry. Obviously, this needed clarification. This sentence now reads:

*The observed asymmetric extent of ice north and south of the Alps can be explained by the modelled transient nature of the LGM extent without involving north-south gradients in temperature and precipitation change (Sect. 4.1).*

Finally, we thank you again very much for the time and effort you put into our manuscript.

**References**

Aschwanden, A., Bueler, E., Khroulev, C., and Blatter, H.: An enthalpy formulation for glaciers and ice sheets, J. Glaciol., 58, 441–457, doi:10.3189/2012JoG11J088, 2012.

Feldmann, J. and Levermann, A.: From cyclic ice streaming to Heinrich-like events: the grow-and-surge instability in the Parallel Ice Sheet Model, The Cryosphere, 11, 1913–1932, doi:10.5194/tc-11-1913-2017, 2017.

Kuchar, J., Milne, G., Hubbard, A., Patton, H., Bradley, S., Shennan, I., and Edwards, R.: Evaluation of a numerical model of the British-Irish ice sheet using relative sea-level data: implications for the interpretation of trimline observations, J. Quaternary Sci., 27, 597–605, doi:10.1002/jqs.2552, 2012.

Mey, J., Scherler, D., Wickert, A. D., Egholm, D. L., Tesauro, M., Schildgen, T. F., and Strecker, M. R.: Glacial isostatic uplift of the European Alps, Nature Communications, 7, 13 382, doi:10.1038/ncomms13382, 2016.

Monegato, G., Ravazzi, C., Donegana, M., Pini, R., Calderoni, G., and Wick, L.: Evidence of a two-fold glacial advance during the last glacial maximum in the Tagliamento end moraine system (eastern Alps), Quaternary Res., 68, 284–302, doi:10.1016/j.yqres.2007.07.002, 2007.

Pelt, W. J. V. and Oerlemans, J.: Numerical simulations of cyclic behaviour in the Parallel Ice Sheet Model (PISM), J. Glaciol., 58, 347–360, doi:10.3189/2012jog11j217, 2012.

Penck, A. and Brückner, E.: Die alpen im Eiszeitalter, Tauchnitz, Leipzig, 1909.

Sevestre, H. and Benn, D. I.: Climatic and geometric controls on the global distribution of surge-type glaciers: implications for a unifying model of surging, J. Glaciol., 61, 646–662, doi:10.3189/2015jog14j136, 2015.

Wirsig, C., Zasadni, J., Christl, M., Akçar, N., and Ivy-Ochs, S.: Dating the onset of LGM ice surface lowering in the High Alps, Quaternary Sci. Rev., 143, 37–50, doi:10.1016/j.quascirev.2016.05.001, https://doi.org/10.1016/j.quascirev.2016.05.001, 2016.

---

## Author Response (AR1)

**Authors' response to the Editor**
J. Seguinot, on behalf of all authors.
July 31, 2018

Dear Andreas Vieli,

We apologize for delays accumulated during the peer review of our manuscript. We believe that we have addressed all points raised by the reviews and hereby submit our revised version to *The Cryosphere*. Please find hereafter a short summary and a marked-up file listing changes made to the manuscript since its first publication in *The Cryosphere Discussion*. Please refer to our public reponses for more detail explanations.

- **Sect. 2 (Ice sheet model set-up):** We have made explicit the lack of measurements on ice softness for water contents above 1 %, clarified that subglacial water is not transported, explained the effect of erroneous elastic thickness, and justified our value for the air temperature lapse-rate.

- **Sect. 3 (Palaeo-climate forcing):** We have provided a short review on available palaeo-climate proxy records in or near the Alps, summarized the state of knowledge on past precipitation changes, and pointed out inconsistencies between our reference climate forcing and other data.

- **Sect. 4 (Results and discussion):** We have pointed out inconsistent LGM ice extent reconstructions in the south-western Alps, removed the discussion on self-sustained ice domes, and discussed uncertainties regarding our conclusion on trimlines. *New references were added following private comments from Christopher Carcaillet and Wilfried Haeberli.*

- **Sect. 5 (Conclusions):** We have added a word of caution regarding our conclusion on EPICA being the optimal forcing, listed major sources of uncertainties, and emphasized the need for more realistic climate forcing in future studies.

- **Fig. 1:** We have removed self-sustained ice domes and updated our somewhat arbitrary choice of transfluence locations.

- **Fig. 6:** We have corrected histogram data and replaced areas modelled to have experienced temperate ice for less than 1 ka by frozen-based areas at the time of maximum ice thicknes.

- **Various:** *We found that thicker 200 m contour lines were disturbing other figure elements and eventually decided to keep original thickness inconsistently with our response to referee #2.* We have implemented numerous suggestions for text improvements.

We thank you again very much for your editorial work on our manuscript.

[revised manuscript text omitted]

---

## Author Response (AR2)

**Authors' response to the Editor**
J. Seguinot, on behalf of all authors.
August 19, 2018

Dear Andreas Vieli,

Thank you very much for supporting our manuscript for publication in *The Cryosphere*. We have adressed your comments and added a few references to the manuscript as detailed below.

**Editor comments**

**p. 1, l. 14-17:** somewhat odd phrasing: '...cause ...to be modelled...', maybe '...produces in the model...' or similar would be better.

Thanks. Done.

**p. 3, l. 4:** 'it still remains incompletely known' is awkward phrasing, maybe better say: '...knowledge remains incomplete...'

We replaced with "uncertainties remain on".

**p. 3, l. 7-8:** the added sentence starting with 'Additional...' seems slightly out of place here (or the flow is someaht disrupted). Maybe better move it to end of nextz paragraph (p. 3, l. 14).

Done.

**p. 3, l. 9:** starting a paragraph with a 'But...' is awkward, maybe change back to 'Here we...' in particular when you move the sentence before.

Done.

**p. 3, l. 32:** would 'different' instead of 'distinct' not be more appropriate.

We replaced with "different".

**p. 4, l. 8:** 'In the SIA, topographic roughness using a bed smoother range of 5 km (Schoof, 2003).' I really do not understand this sentence (what is message), nor its connection to the sentence before.

Sorry, two words were missing here: "topographic roughness is parametrized using a bed smoother range of 5 km." We have moved this sentence directly after the previous mention of the SIA.

**p. 4, l. 24:** '...without horizontal transport...'?

Done.

**p. 4, l. 25:** maybe simplify to '...is assumed to drain instantaneously and is removed....'

Done.

**p. 7, l. 3:** 'above average' of what? What is reference?

We have changed to "above annual lapse rates measured in the Alps". The reference describes lapse rate measurements.

**p. 14, fig. 4:** are the surface contours really in 200m intervals?, that would make the ice surface only a bit over 1200m high. Or do you leave some out? You should add some labels to the surface contour lines so one can judge the surface elevation better.

An incorrect version of the figure missing 200m contours was uploaded. This has been corrected. Thank you very much for noticing.

**p. 13, l. 30 – p. 14, l. 3:** but jouvet (TC) claims (based on tracking erratics) that north south contrast in in precip is needed. Is this still consistent with this work? Maybe should be included. This is mentioned on next page but would be relevant here as well.

A reference to (Jouvet et al., 2017) was added. However, to avoid redundancy with the next section on flow patterns, we would prefer to limit the discussion in this section to modelled ice marginal positions.

**p. 14, fig. 4 caption:** for '(b)' it should say what the vertical black line (time of max extent in (a) at 24.57 ka???) refers to.

We have made explicit that: "The black vertical line indicates the modelled age of maximum ice cover at 24.57 ka."

**p. 15, l. 31:** 'simulation' (not sumulation)

Done.

**p. 16, l. 12:** 'Fig. 5' has nothing to do with 'reviews', what do you mean with 'Fig. 5 for reviews'???

With "cf. Ivy-Ochs, 2015; Wirsig et al., 2016, Fig. 5 for reviews", we meant to refer to Fig. 5 in the paper by (Wirsig et al., 2016). We have removed "Fig. 5" in this particular instance.

**p. 19, l. 15-19:** in this context of sensitivity to basal sliding parameter (and lacking exploration) maybe some reference to the Phd-thesis of Becker which includes at least some preliminary (mostly steady state) sliding investigation could be made.

Done.

**p. 21, l. 19:** 'last major Alpine reliefs' ...? Do you mean the 'last foothills' of the alps? Clarify.

We have replaced this formulation with "the outermost limestone reliefs".

**p. 23, l. 1:** is it not rather 'glacier physics' as the till is not ice (or ice and glacier physics).

Correct. We have changed to "glacier physics".

**New references**

References to several new papers were added (Cohen et al., 2018; Gild et al., 2018; Ivy-Ochs et al., 2018; Barrett et al., 2018; Spötl et al., 2018).

We are currently applying for data publication DOIs. We thank you again very much for your editorial work on our manuscript.

**References**

[revised manuscript text omitted]